# Revisiting Smoothed Online Learning

**Lijun Zhang**[1,2], **Wei Jiang**[1], **Shiyin Lu**[1], **Tianbao Yang**[3]

[1]National Key Laboratory for Novel Software Technology, Nanjing University, Nanjing, China
[2]Peng Cheng Laboratory, Shenzhen, Guangdong, China
[3]Department of Computer Science, The University of Iowa, Iowa City, IA 52242, USA
{zhanglj, jiangw, lusy}@lamda.nju.edu.cn, tianbao-yang@uiowa.edu

## Abstract

In this paper, we revisit the problem of smoothed online learning, in which the online learner suffers both a hitting cost and a switching cost, and target two performance metrics: competitive ratio and dynamic regret with switching cost. To bound the competitive ratio, we assume the hitting cost is known to the learner in each round, and investigate the simple idea of balancing the two costs by an optimization problem. Surprisingly, we find that minimizing the hitting cost alone is $\max(1, \frac{2}{\alpha})$-competitive for $\alpha$-polyhedral functions and $1 + \frac{4}{\lambda}$-competitive for $\lambda$-quadratic growth functions, both of which improve state-of-the-art results significantly. Moreover, when the hitting cost is both convex and $\lambda$-quadratic growth, we reduce the competitive ratio to $1 + \frac{2}{\sqrt{\lambda}}$ by minimizing the weighted sum of the hitting cost and the switching cost. To bound the dynamic regret with switching cost, we follow the standard setting of online convex optimization, in which the hitting cost is convex but hidden from the learner before making predictions. We modify Ader, an existing algorithm designed for dynamic regret, slightly to take into account the switching cost when measuring the performance. The proposed algorithm, named as Smoothed Ader, attains an optimal $O(\sqrt{T(1 + P_T)})$ bound for dynamic regret with switching cost, where $P_T$ is the path-length of the comparator sequence. Furthermore, if the hitting cost is accessible in the beginning of each round, we obtain a similar guarantee without the bounded gradient condition, and establish an $\Omega(\sqrt{T(1 + P_T)})$ lower bound to confirm the optimality.

## 1 Introduction

Online learning is the process of making a sequence of predictions given knowledge of the answer to previous tasks and possibly additional information [Shalev-Shwartz, 2011]. While the traditional online learning aims to make the prediction as accurate as possible, in this paper, we study smoothed online learning (SOL), where the online learner incurs a switching cost for changing its predictions between rounds [Cesa-Bianchi et al., 2013]. SOL has received lots of attention recently because in many real-world applications, a change of action usually brings some additional cost. Examples include the dynamic right-sizing for data centers [Lin et al., 2011], geographical load balancing [Lin et al., 2012], real-time electricity pricing [Kim and Giannakis, 2014], video streaming [Joseph and de Veciana, 2012], spatiotemporal sequence prediction [Kim et al., 2015], multi-timescale control [Goel et al., 2017], and thermal management [Zanini et al., 2010].

Specifically, SOL is performed in a sequence of consecutive rounds, where at round $t$ the learner is asked to select a point $\mathbf{x}_t$ from the decision set $\mathcal{X}$, and suffers a hitting cost $f_t(\mathbf{x}_t)$. Depending on the performance metric, the learner *may* be allowed to observe $f_t(\cdot)$ when making decisions, which is different from the traditional online learning in which $f_t(\cdot)$ is revealed to the learner after submitting the decision [Cesa-Bianchi and Lugosi, 2006]. Additionally, the learner also incurs a switching cost $m(\mathbf{x}_t, \mathbf{x}_{t-1})$ for changing decisions between successive rounds. The switching cost $m(\mathbf{x}_t, \mathbf{x}_{t-1})$

35th Conference on Neural Information Processing Systems (NeurIPS 2021).

could be any distance function, such as the $\ell_2$-norm distance $\|\mathbf{x}_t - \mathbf{x}_{t-1}\|$ and the squared $\ell_2$-norm distance $\|\mathbf{x}_t - \mathbf{x}_{t-1}\|^2/2$ [Goel et al., 2019]. In the literature, there are two performance metrics for SOL: competitive ratio and dynamic regret with switching cost.

Competitive ratio is popular in the community of online algorithms [Borodin and El-Yaniv, 1998]. It is defined as the worst-case ratio of the total cost incurred by the online learner and the offline optimal cost:

$$\frac{\sum_{t=1}^{T} \left( f_t(\mathbf{x}_t) + m(\mathbf{x}_t, \mathbf{x}_{t-1}) \right)}{\min_{\mathbf{u}_0, \mathbf{u}_1, \ldots, \mathbf{u}_T \in \mathcal{X}} \sum_{t=1}^{T} \left( f_t(\mathbf{u}_t) + m(\mathbf{u}_t, \mathbf{u}_{t-1}) \right)}. \tag{1}$$

When focusing on the competitive ratio, the learner can observe $f_t(\cdot)$ before picking $\mathbf{x}_t$. The problem is still nontrivial due to the coupling created by the switching cost. On the other hand, dynamic regret with switching cost is a generalization of dynamic regret—a popular performance metric in the community of online learning [Zinkevich, 2003]. It is defined as the difference between the total cost incurred by the online learner and that of an arbitrary comparator sequence $\mathbf{u}_0, \mathbf{u}_1, \ldots, \mathbf{u}_T \in \mathcal{X}$:

$$\sum_{t=1}^{T} \left( f_t(\mathbf{x}_t) + m(\mathbf{x}_t, \mathbf{x}_{t-1}) \right) - \sum_{t=1}^{T} \left( f_t(\mathbf{u}_t) + m(\mathbf{u}_t, \mathbf{u}_{t-1}) \right). \tag{2}$$

Different from previous work [Chen et al., 2018, Goel et al., 2019], we did not introduce the minimization operation over $\mathbf{u}_0, \mathbf{u}_1, \ldots, \mathbf{u}_T$ in (2). The reason is that we want to bound (2) by certain regularities of the comparator sequence, such as the path-length

$$P_T(\mathbf{u}_0, \mathbf{u}_1, \ldots, \mathbf{u}_T) = \sum_{t=1}^{T} \|\mathbf{u}_t - \mathbf{u}_{t-1}\|. \tag{3}$$

When focusing on (2), $f_t(\cdot)$ is generally hidden from the learner before submitting $\mathbf{x}_t$. The conditions for bounding the two metrics are very different, so we study competitive ratio and dynamic regret with switching cost separately. To bound the two metrics simultaneously, we refer to Andrew et al. [2013] and Daniely and Mansour [2019], especially the meta-algorithm in the latter work.

This paper follows the line of research stemmed from online balanced descent (OBD) [Chen et al., 2018, Goel and Wierman, 2019]. The key idea of OBD is to find an appropriate balance between the hitting cost and the switching cost through iterative projections. It has been shown that OBD and its variants are able to exploit the analytical properties of the hitting cost (e.g., polyhedral, strongly convex) to derive dimension-free competitive ratio. At this point, it would be natural to ask why not use the *greedy* algorithm, which minimizes the weighted sum of the hitting cost and the switching cost in each round, i.e.,

$$\min_{\mathbf{x} \in \mathcal{X}} \quad f_t(\mathbf{x}) + \gamma m(\mathbf{x}, \mathbf{x}_{t-1}) \tag{4}$$

to balance the two costs, where $\gamma \geq 0$ is the trade-off parameter. We note that the greedy algorithm is usually treated as the baseline in competitive analysis [Borodin and El-Yaniv, 1998], but its usage for smoothed online learning is quite limited. One result is given by Goel et al. [2019], who demonstrate that the greedy algorithm as a special case of Regularized OBD (R-OBD), is optimal for strongly convex functions. Besides, Lin et al. [2020] have analyzed the greedy algorithm with $\gamma = 0$, named as the *naive* approach below, for polyhedral functions and quadratic growth functions.

In this paper, we make the following contributions towards understanding the greedy algorithm.

- For $\alpha$-polyhedral functions, the competitive ratio of the naive approach is $\max(1, \frac{2}{\alpha})$, which is a significant improvement over the $3 + \frac{8}{\alpha}$ competitive ratio of OBD [Chen et al., 2018] and the $1 + \frac{2}{\alpha}$ ratio proved by Lin et al. [2020, Lemma 1]. When $\alpha > 2$, the ratio becomes 1, indicating that the naive approach is optimal in this scenario.
- For $\lambda$-quadratic growth functions, the competitive ratio of the naive algorithm is $1 + \frac{4}{\lambda}$, which matches the lower bound of this algorithm [Goel et al., 2019, Theorem 5], and is better than the $\max(1 + \frac{6}{\lambda}, 4)$ ratio obtained by Lin et al. [2020, Lemma 1].
- If the hitting cost is both convex and $\lambda$-quadratic growth, the greedy algorithm with $\gamma > 0$ attains a $1 + \frac{2}{\sqrt{\lambda}}$ competitive ratio, which demonstrates the advantage of taking the switching cost into considerations. Our $1 + \frac{2}{\sqrt{\lambda}}$ ratio is on the same order as Greedy OBD [Goel et al., 2019, Theorem 3] but with much smaller constants.

- Our analysis of the naive approach and the greedy algorithm is very simple. In contrast, both OBD and Greedy OBD rely on intricate geometric arguments.

While both OBD and R-OBD are equipped with sublinear dynamic regret with switching cost, they are unsatisfactory in the following aspects:

- The regret of OBD depends on an upper bound of the path-length instead of the path-length itself [Chen et al., 2018, Corollary 11], making it nonadaptive.
- The regret of R-OBD is adaptive but it uses the squared $\ell_2$-norm to measure the switching cost, which may not be suitable for general convex functions [Goel et al., 2019].[1]
- Both OBD and R-OBD observe $f_t(\cdot)$ before selecting $\mathbf{x}_t$, which violates the convention of online learning.

To avoid the above limitations, we demonstrate that a small change of Ader [Zhang et al., 2018a], which is an existing algorithm designed for dynamic regret, is sufficient to minimize the dynamic regret with switching cost under the setting of online convex optimization [Shalev-Shwartz, 2011]. Ader runs multiple online gradient descent (OGD) [Zinkevich, 2003] with different step sizes as expert-algorithms, and uses Hedge [Freund and Schapire, 1997] as the meta-algorithm to aggregate predictions from experts. The only modification is to incorporate the switching cost into the loss of Hedge. The proposed algorithm, named as Smoothed Ader (SAder), attains the optimal $O(\sqrt{T(1 + P_T)})$ dynamic regret, where $P_T$ is the path-length defined in (3). Thus, our regret bound is *adaptive* because it automatically becomes small when the comparators change slowly. Finally, we also investigate the case that the hitting cost is available before predictions, and establish a similar result without the bounded gradient condition. To this end, we design a *lookahead* version of SAder, which chooses the greedy algorithm in (4) as the expert and utilizes the cost of the current round in Hedge. To show the optimality of this algorithm, we further establish an $\Omega(\sqrt{T(1 + P_T)})$ lower bound under the lookahead setting.

## 2 Related work

This section reviews related work on smoothed online learning (SOL) and dynamic regret.

### 2.1 Smoothed online learning

SOL has been investigated under the setting of multi-armed bandits [Agrawal et al., 1990, Guha and Munagala, 2009, Dekel et al., 2014, Koren et al., 2017a,b], prediction with expert advice [Cesa-Bianchi et al., 2013], and online convex optimization [Lin et al., 2011, Bansal et al., 2015, Zhao et al., 2020b]. In the following, we mainly discuss smoothed online convex optimization (SOCO).

The early works on SOCO focus on designing competitive algorithms in the low-dimensional setting [Lin et al., 2011]. In particular, Bansal et al. [2015] show that for SOCO on the real line, the competitive ratio can be upper bounded by 2, which is proved to be optimal [Antoniadis and Schewior, 2018]. They also establish a competitive ratio of 3 under the memoryless setting. In the study of SOCO, it is common to assume that the learner has access to predictions of future hitting costs, and several algorithms [Lin et al., 2012, Chen et al., 2015, 2016, Li et al., 2018, Li and Li, 2020] have been developed based on receding horizon control (RHC) [Kwon and Han, 2005]. In fact, the greedy algorithm in (4) can be treated as a variant of RHC. However, previous results for RHC are limited to special problems, and (4) remains under-explored. For example, when the learner can observe the next $W$ hitting costs, Li et al. [2018] demonstrate that both the competitive ratio and the dynamic regret with switching cost decay exponentially fast with $W$. But their analysis relies on very strong conditions, including strong convexity and smoothness.

One milestone is the online balanced descent (OBD) [Chen et al., 2018], which has dimension-free competitive ratio even when the learner can only observe the hitting cost of the current round. Specifically, OBD iteratively projects the previous point onto a carefully chosen level set of the hitting cost so as to balance the switching cost and the hitting cost. When the hitting cost is $\alpha$-polyhedral and convex, and the switching cost is the $\ell_2$-norm distance, OBD attains a $3 + \frac{8}{\alpha}$ competitive ratio. Furthermore, OBD can also be tuned to control the dynamic regret with switching cost. Let $L$ be

---

[1] We usually assume the convex function is Lipschitz continuous, and in this case choosing the $\ell_2$-norm distance as the switching cost makes it on the same order as the hitting cost.

an upper bound of the path-length of the comparator sequence, i.e., $P_T(\mathbf{u}_0, \mathbf{u}_1, \ldots, \mathbf{u}_T) \le L$. Chen et al. [2018, Corollary 11] have proved that

$$\sum_{t=1}^{T} \left( f_t(\mathbf{x}_t) + \|\mathbf{x}_t - \mathbf{x}_{t-1}\| \right) - \min_{P_T(\mathbf{u}_0, \mathbf{u}_1, \ldots, \mathbf{u}_T) \le L} \sum_{t=1}^{T} \left( f_t(\mathbf{u}_t) + \|\mathbf{u}_t - \mathbf{u}_{t-1}\| \right) = O\left( \sqrt{TL} \right) \quad (5)$$

leading to sublinear regret when $L = o(T)$. However, the upper bound is nonadaptive because it depends on $L$ instead of the actual path-length $P_T$.

Later, Goel and Wierman [2019] demonstrate that OBD is $3 + O(\frac{1}{\lambda})$-competitive for $\lambda$-strongly convex functions, when the switching cost is set to be the squared $\ell_2$-norm distance. In a subsequent work, Goel et al. [2019, Theorem 4] propose Regularized OBD (R-OBD), which improves the competitive ratio to $\frac{1}{2} + \frac{1}{2}\sqrt{1 + \frac{4}{\lambda}}$, matching the lower bound of strongly convex functions exactly [Goel et al., 2019, Theorem 1]. R-OBD includes (4) as a special case, which also enjoys the optimal competitive ratio for strongly convex functions. Furthermore, Goel et al. [2019, Theorem 6] have analyzed the dynamic regret of R-OBD and the following result can be extracted from that paper

$$\sum_{t=1}^{T} \left( f_t(\mathbf{x}_t) + \frac{1}{2}\|\mathbf{x}_t - \mathbf{x}_{t-1}\|^2 \right) - \sum_{t=1}^{T} \left( f_t(\mathbf{u}_t) + \frac{1}{2}\|\mathbf{u}_t - \mathbf{u}_{t-1}\|^2 \right) = O\left( \sqrt{T \sum_{t=1}^{T} \|\mathbf{u}_t - \mathbf{u}_{t-1}\|^2} \right)$$

Compared with (5), this bound is adaptive because the upper bound depends on the switching cost of the comparator sequence. However, it chooses the squared $\ell_2$-norm as the switching cost, which may not be suitable for general convex functions. When the hitting cost is both quasiconvex and $\lambda$-quadratic growth, Goel et al. [2019, Theorem 3] have demonstrated that their Greedy OBD algorithm attains an $O(1/\sqrt{\lambda})$ competitive ratio, as $\lambda \to 0$.

Lin et al. [2020] have analyzed the naive approach which ignores the switching cost and simply minimizes the hitting cost in each round, i.e., the greedy algorithm with $\gamma = 0$. It is a bit surprising that this naive approach is $1 + \frac{2}{\alpha}$-competitive for $\alpha$-polyhedral functions and $\max(1 + \frac{6}{\lambda}, 4)$-competitive for $\lambda$-quadratic growth functions, without any convexity assumption [Lin et al., 2020, Lemma 1]. Argue et al. [2020a] have investigated a hybrid setting in which the hitting cost is both $\lambda$-strongly convex and $H$-smooth, but the switching cost is the $\ell_2$-norm distance instead of the squared one. They develop Constrained OBD, and establish a $4 + 4\sqrt{2H/\lambda}$ competitive ratio. However, their analysis relies on a strong condition that the hitting cost is non-negative over the whole space, i.e., $\min_{\mathbf{x} \in \mathbb{R}^d} f_t(\mathbf{x}) = 0$, as opposed to the usual condition $\min_{\mathbf{x} \in \mathcal{X}} f_t(\mathbf{x}) = 0$.

Finally, we note that SOCO is closely related to convex body chasing (CBC) [Friedman and Linial, 1993, Antoniadis et al., 2016, Bansal et al., 2018, Argue et al., 2019, Bubeck et al., 2019, 2020]. In this problem, the online learner receives a sequence of convex bodies $\mathcal{X}_1, \ldots, \mathcal{X}_T \subseteq \mathbb{R}^d$ and must select one point from each body, and the goal is to minimize the total movement between consecutive output points. Apparently, we can treat CBC as a special case of SOCO by defining the hitting cost $f_t(\cdot)$ as the indicator function of $\mathcal{X}_t$, which means that the domains of hitting costs are allowed to be different. On the other hand, we can also formulate a $d$-dimensional SOCO problem as a $d + 1$-dimensional CBC problem [Lin et al., 2020, Proposition 1]. For the general setting of CBC, the competitive ratio exhibits a polynomial dependence on the dimensionality, and the state-of-the-art result is $O(\min(d, \sqrt{d \log T}))$ [Argue et al., 2020b, Sellke, 2020], which nearly match the $\Omega(\sqrt{d})$ lower bound [Friedman and Linial, 1993]. Our paper aims to derive dimensionality-independent competitive ratios and sublinear dynamic regret for SOCO, under appropriate conditions.

## 2.2 Dynamic regret

Recently, dynamic regret has attained considerable interest in the community of online learning [Zhang, 2020]. The motivation of dynamic regret is to deal with changing environments, in which the optimal decision may change over time. It is defined as the difference between the cumulative loss of the learner and that of a sequence of comparators $\mathbf{u}_1, \ldots, \mathbf{u}_T \in \mathcal{X}$:

$$\text{D-Regret}(\mathbf{u}_1, \ldots, \mathbf{u}_T) = \sum_{t=1}^{T} f_t(\mathbf{x}_t) - \sum_{t=1}^{T} f_t(\mathbf{u}_t). \quad (6)$$

In the general form of dynamic regret, $\mathbf{u}_1, \ldots, \mathbf{u}_T$ could be an *arbitrary* sequence [Zinkevich, 2003, Hall and Willett, 2013, Zhang et al., 2018a, 2020, Cutkosky, 2020, Zhao et al., 2020a], and in the restricted form, they are chosen as the minimizers of online functions, i.e., $\mathbf{u}_t \in \operatorname{argmin}_{\mathbf{x} \in \mathcal{X}} f_t(\mathbf{x})$ [Jadbabaie et al., 2015, Besbes et al., 2015, Yang et al., 2016, Mokhtari et al., 2016, Zhang et al., 2017, 2018b, Wan et al., 2021, Zhao and Zhang, 2021]. While it is well-known that sublinear dynamic regret is unattainable in the worst case, one can bound the dynamic regret in terms of some regularities of the comparator sequence. An instance is given by Zinkevich [2003], who introduces the notion of path-length defined in (3) to measure the temporal variability of the comparator sequence, and derives an $O(\sqrt{T}(1 + P_T))$ bound for the dynamic regret of OGD. Later, Zhang et al. [2018a] develop adaptive learning for dynamic environment (Ader), which achieves the optimal $O(\sqrt{T(1 + P_T)})$ dynamic regret. In this paper, we show that a small change of Ader attains the same bound for dynamic regret with switching cost.

## 3 Competitive ratio

In this section, we focus on competitive ratio. Without loss of generality, we assume the hitting cost is non-negative, since the competitive ratio can only improve if this is not the case.

### 3.1 Polyhedral functions

We first introduce the definition of polyhedral functions.

**Definition 1** *A function* $f(\cdot) : \mathcal{X} \mapsto \mathbb{R}$ *with minimizer* $\mathbf{v}$ *is* $\alpha$*-polyhedral if*

$$f(\mathbf{x}) - f(\mathbf{v}) \geq \alpha \|\mathbf{x} - \mathbf{v}\|, \ \forall \mathbf{x} \in \mathcal{X}. \tag{7}$$

We note that polyhedral functions have been used for stochastic network optimization [Huang and Neely, 2011] and geographical load balancing [Lin et al., 2012].

Following Chen et al. [2018], we set the switching cost as $m(\mathbf{x}_t, \mathbf{x}_{t-1}) = \|\mathbf{x}_t - \mathbf{x}_{t-1}\|$. Intuitively, we may expect that the switching cost should be taken into consideration when making decisions. However, our analysis shows that minimizing the hitting cost alone yields the tightest competitive ratio so far. Specifically, we consider the following naive approach that ignores the switching cost and selects

$$\mathbf{x}_t = \operatorname*{argmin}_{\mathbf{x} \in \mathcal{X}} f_t(\mathbf{x}). \tag{8}$$

The theoretical guarantee of (8) is stated below.

**Theorem 1** *Suppose each* $f_t(\cdot) : \mathcal{X} \mapsto \mathbb{R}$ *with minimizer* $\mathbf{v}_t$ *is* $\alpha$*-polyhedral. We have*

$$\sum_{t=1}^{T} \left( f_t(\mathbf{x}_t) + \|\mathbf{x}_t - \mathbf{x}_{t-1}\| \right) \leq \max\left(1, \frac{2}{\alpha}\right) \sum_{t=1}^{T} \left( f_t(\mathbf{u}_t) + \|\mathbf{u}_t - \mathbf{u}_{t-1}\| \right), \ \forall \mathbf{u}_0, \mathbf{u}_1, \ldots, \mathbf{u}_T \in \mathcal{X}$$

*where we assume* $\mathbf{x}_0 = \mathbf{u}_0$.

**Remark:** Our $\max(1, \frac{2}{\alpha})$ competitive ratio is much better than the $3 + \frac{8}{\alpha}$ ratio of OBD [Chen et al., 2018], and also better than the $1 + \frac{2}{\alpha}$ ratio established by Lin et al. [2020] for (8). When $\alpha > 2$, the ratio becomes 1, indicating that the naive approach is optimal in this scenario. Furthermore, the proof of Theorem 1 is much simpler than that of OBD, and refines that of Lin et al. [2020, Lemma 1].

We have analyzed the greedy algorithm (4) with $\gamma > 0$, but the competitive ratio does not improve. So, the current knowledge suggests that there is no need to consider the switching cost when facing polyhedral functions. It is unclear whether this is an artifact of our analysis or an inherent property, and will be investigated in the future. Due to space limitations, all the proofs are deferred to the supplementary.

### 3.2 Quadratic growth functions

In this section, we consider the quadratic growth condition.

**Definition 2** *A function $f(\cdot) : \mathcal{X} \mapsto \mathbb{R}$ with minimizer $\mathbf{v}$ is $\lambda$-quadratic growth if*

$$f(\mathbf{x}) - f(\mathbf{v}) \geq \frac{\lambda}{2}\|\mathbf{x} - \mathbf{v}\|^2, \ \forall \mathbf{x} \in \mathcal{X}. \tag{9}$$

The quadratic growth condition has been exploited by the optimization community [Drusvyatskiy and Lewis, 2018, Necoara et al., 2019] to establish linear convergence, and this condition is weaker than strong convexity [Hazan and Kale, 2011].

Following Goel et al. [2019], we set the switching cost as $m(\mathbf{x}_t, \mathbf{x}_{t-1}) = \|\mathbf{x}_t - \mathbf{x}_{t-1}\|^2/2$. We also consider the naive approach in (8) and have the following theoretical guarantee.

**Theorem 2** *Suppose each $f_t(\cdot) : \mathcal{X} \mapsto \mathbb{R}$ with minimizer $\mathbf{v}_t$ is $\lambda$-quadratic growth. We have*

$$\sum_{t=1}^{T} \left( f_t(\mathbf{x}_t) + \frac{1}{2}\|\mathbf{x}_t - \mathbf{x}_{t-1}\|^2 \right) \leq \left(1 + \frac{4}{\lambda}\right) \sum_{t=1}^{T} \left( f_t(\mathbf{u}_t) + \frac{1}{2}\|\mathbf{u}_t - \mathbf{u}_{t-1}\|^2 \right),$$

*for all $\mathbf{u}_0, \mathbf{u}_1, \ldots, \mathbf{u}_T \in \mathcal{X}$, where we assume $\mathbf{x}_0 = \mathbf{u}_0$.*

**Remark:** The above theorem implies that the naive approach achieves a competitive ratio of $1 + \frac{4}{\lambda}$, which matches the lower bound of this algorithm [Goel et al., 2019, Theorem 5]. Furthermore, it is also much better than the $\max(1 + \frac{6}{\lambda}, 4)$ ratio established by Lin et al. [2020] for (8). Similar to the case of polyhedral functions, it seems safe to ignore the switching cost here.

### 3.3 Convex and quadratic growth functions

When $f_t(\cdot)$ is both quasiconvex and $\lambda$-quadratic growth, Goel et al. [2019] have established an $O(1/\sqrt{\lambda})$ competitive ratio for Greedy OBD. Inspired by this result, we introduce convexity to further improve the competitive ratio. In this case, the switching cost plays a role in deriving tighter competitive ratios. Specifically, we choose the greedy algorithm with $\gamma > 0$ to select $\mathbf{x}_t$, i.e.,

$$\mathbf{x}_t = \operatorname*{argmin}_{\mathbf{x} \in \mathcal{X}} \left( f_t(\mathbf{x}) + \frac{\gamma}{2}\|\mathbf{x} - \mathbf{x}_{t-1}\|^2 \right). \tag{10}$$

The theoretical guarantee of (10) is stated below.

**Theorem 3** *Suppose the domain $\mathcal{X}$ is convex, and each $f_t(\cdot) : \mathcal{X} \mapsto \mathbb{R}$ with minimizer $\mathbf{v}_t$ is $\lambda$-quadratic growth and convex. By setting $\gamma = \lambda/(\lambda + \sqrt{\lambda})$, we have*

$$\sum_{t=1}^{T} \left( f_t(\mathbf{x}_t) + \frac{1}{2}\|\mathbf{x}_t - \mathbf{x}_{t-1}\|^2 \right) \leq \left(1 + \frac{2}{\sqrt{\lambda}}\right) \sum_{t=1}^{T} \left( f_t(\mathbf{u}_t) + \frac{1}{2}\|\mathbf{u}_t - \mathbf{u}_{t-1}\|^2 \right),$$

*for all $\mathbf{u}_0, \mathbf{u}_1, \ldots, \mathbf{u}_T \in \mathcal{X}$, where we assume $\mathbf{x}_0 = \mathbf{u}_0$.*

**Remark:** The above theorem shows that the competitive ratio is improved to $1 + \frac{2}{\sqrt{\lambda}}$ under the additional convexity condition. According to the lower bound of strongly convex functions [Goel et al., 2019, Theorem 1], the ratio in Theorem 3 is optimal up to constant factors. Compared with Greedy OBD [Goel et al., 2019, Theorem 3], our assumption is slightly stronger, since we require convexity instead of quasiconvexity. However, our algorithm and analysis are much simpler, and the constants in our bound are much smaller.

## 4 Dynamic regret with switching cost

When considering dynamic regret with switching cost, we adopt the common assumptions of online convex optimization (OCO) [Shalev-Shwartz, 2011].

**Assumption 1** *All the functions $f_t$'s are convex over their domain $\mathcal{X}$.*

**Assumption 2** *The gradients of all functions are bounded by $G$, i.e.,*

$$\max_{\mathbf{x} \in \mathcal{X}} \|\nabla f_t(\mathbf{x})\| \leq G, \ \forall t \in [T]. \tag{11}$$

---
**Algorithm 1** SAder: Meta-algorithm
---
**Require:** A step size $\beta$, and a set $\mathcal{H}$ containing step sizes for experts
 1: Activate a set of experts $\{E^\eta | \eta \in \mathcal{H}\}$ by invoking the expert-algorithm for each step size $\eta \in \mathcal{H}$
 2: Sort step sizes in ascending order $\eta_1 \leq \eta_2 \leq \cdots \leq \eta_N$, and set $w_1^{\eta_i} = \frac{C}{i(i+1)}$
 3: **for** $t = 1, \ldots, T$ **do**
 4:     Receive $\mathbf{x}_t^\eta$ from each expert $E^\eta$
 5:     Output the weighted average $\mathbf{x}_t = \sum_{\eta \in \mathcal{H}} w_t^\eta \mathbf{x}_t^\eta$
 6:     Observe the loss function $f_t(\cdot)$
 7:     Update the weight of each expert by (14)
 8:     Send gradient $\nabla f_t(\mathbf{x}_t)$ to each expert $E^\eta$
 9: **end for**
---

**Assumption 3** *The diameter of the domain $\mathcal{X}$ is bounded by $D$, i.e.,*

$$\max_{\mathbf{x}, \mathbf{x}' \in \mathcal{X}} \|\mathbf{x} - \mathbf{x}'\| \leq D. \tag{12}$$

Assumption 2 implies that the hitting cost is Lipschitz continuous, so it is natural to set the switching cost as $m(\mathbf{x}_t, \mathbf{x}_{t-1}) = \|\mathbf{x}_t - \mathbf{x}_{t-1}\|$.

### 4.1 The standard setting

We first follow the standard setting of OCO in which the learner can not observe the hitting cost when making predictions, and develop an algorithm based on Ader [Zhang et al., 2018a]. Specifically, we demonstrate that a small change of Ader, which modifies the loss of the meta-algorithm to take into account the switching cost of experts, is sufficient to minimize the dynamic regret with switching cost. Our proposed method is named as Smoothed Ader (SAder), and stated below.[2]

**Meta-algorithm** The meta-algorithm is similar to that of Ader [Zhang et al., 2018a, Algorithm 3], and summarized in Algorithm 1. The inputs of the meta-algorithm are its own step size $\beta$, and a set $\mathcal{H}$ of step sizes for experts. In Step 1, we active a set of experts $\{E^\eta | \eta \in \mathcal{H}\}$ by invoking the expert-algorithm for each $\eta \in \mathcal{H}$. In Step 2, we set the initial weight of each expert. Let $\eta_i$ be the $i$-th smallest step size in $\mathcal{H}$. The weight of $E^{\eta_i}$ is chosen as

$$w_1^{\eta_i} = \frac{C}{i(i+1)}, \text{ and } C = 1 + \frac{1}{|\mathcal{H}|}. \tag{13}$$

In each round, the meta-algorithm receives a set of predictions $\{\mathbf{x}_t^\eta | \eta \in \mathcal{H}\}$ from all experts (Step 4), and outputs the weighted average (Step 5):

$$\mathbf{x}_t = \sum_{\eta \in \mathcal{H}} w_t^\eta \mathbf{x}_t^\eta$$

where $w_t^\eta$ is the weight assigned to expert $E^\eta$. After observing the loss function, the weights of experts are updated according to the exponential weighting scheme (Step 7) [Cesa-Bianchi and Lugosi, 2006]:

$$w_{t+1}^\eta = \frac{w_t^\eta e^{-\beta \ell_t(\mathbf{x}_t^\eta)}}{\sum_{\eta \in \mathcal{H}} w_t^\eta e^{-\beta \ell_t(\mathbf{x}_t^\eta)}} \tag{14}$$

where

$$\ell_t(\mathbf{x}_t^\eta) = \langle \nabla f_t(\mathbf{x}_t), \mathbf{x}_t^\eta - \mathbf{x}_t \rangle + \|\mathbf{x}_t^\eta - \mathbf{x}_{t-1}^\eta\|. \tag{15}$$

When $t = 1$, we set $\mathbf{x}_0^\eta = 0$, for all $\eta \in \mathcal{H}$. As can be seen from (15), we incorporate the switching cost $\|\mathbf{x}_t^\eta - \mathbf{x}_{t-1}^\eta\|$ of expert $E^\eta$ to measure its performance. This is the *only* modification made to Ader. In the last step, we send the gradient $\nabla f_t(\mathbf{x}_t)$ to each expert $E^\eta$ so that they can update their own predictions.

---

[2]In a concurrent work, Zhao et al. [2021] independently develop a similar algorithm for OCO with memory.

---

**Algorithm 2** SAder: Expert-algorithm

---

**Require:** The step size $\eta$
 1: Let $\mathbf{x}_1^\eta$ be any point in $\mathcal{X}$
 2: **for** $t = 1, \ldots, T$ **do**
 3:    Submit $\mathbf{x}_t^\eta$ to the meta-algorithm
 4:    Receive gradient $\nabla f_t(\mathbf{x}_t)$ from the meta-algorithm
 5:
$$\mathbf{x}_{t+1}^\eta = \Pi_{\mathcal{X}}\left[\mathbf{x}_t^\eta - \eta \nabla f_t(\mathbf{x}_t)\right]$$

 6: **end for**

---

**Expert-algorithm**  The expert-algorithm is the same as that of Ader [Zhang et al., 2018a, Algorithm 4], which is OGD over the linearized loss or the surrogate loss

$$s_t(\mathbf{x}) = \langle \nabla f_t(\mathbf{x}_t), \mathbf{x} - \mathbf{x}_t \rangle. \tag{16}$$

For the sake of completeness, we present its procedure in Algorithm 2. The input of the expert is its step size $\eta$. In Step 3 of Algorithm 2, each expert submits its prediction $\mathbf{x}_t^\eta$ to the meta-algorithm, and receives the gradient $\nabla f_t(\mathbf{x}_t)$ in Step 4. Then, in Step 5, it performs gradient descent

$$\mathbf{x}_{t+1}^\eta = \Pi_{\mathcal{X}}\left[\mathbf{x}_t^\eta - \eta \nabla f_t(\mathbf{x}_t)\right]$$

to get the prediction for the next round. Here, $\Pi_{\mathcal{X}}[\cdot]$ denotes the projection onto the nearest point in $\mathcal{X}$.

We have the following theoretical guarantee.

**Theorem 4** *Set*

$$\mathcal{H} = \left\{ \eta_i = 2^{i-1} \sqrt{\frac{D^2}{T(G^2 + 2G)}} \middle| i = 1, \ldots, N \right\} \tag{17}$$

*where*

$$N = \left\lceil \frac{1}{2} \log_2(1 + 2T) \right\rceil + 1, \text{ and } \beta = \frac{2}{(2G+1)D}\sqrt{\frac{2}{5T}}$$

*in Algorithm 1. Under Assumptions 1, 2 and 3, for* any *comparator sequence* $\mathbf{u}_0, \mathbf{u}_1, \ldots, \mathbf{u}_T \in \mathcal{X}$, *SAder satisfies*

$$\sum_{t=1}^{T} \left( f_t(\mathbf{x}_t) + \|\mathbf{x}_t - \mathbf{x}_{t-1}\| \right) - \sum_{t=1}^{T} f_t(\mathbf{u}_t) \tag{18}$$

$$\leq \frac{3}{2}\sqrt{T(G^2 + 2G)\left(D^2 + 2D\sum_{t=1}^{T}\|\mathbf{u}_t - \mathbf{u}_{t-1}\|\right)} + (2G+1)D\sqrt{\frac{5T}{8}}\left[1 + 2\ln(k+1)\right]$$

$$= O\left(\sqrt{T(1+P_T)} + \sqrt{T}(1 + \log\log P_T)\right) = O\left(\sqrt{T(1+P_T)}\right)$$

*where we define* $\mathbf{x}_0 = 0$, *and*

$$k = \left\lfloor \frac{1}{2}\log_2\left(1 + \frac{2P_T}{D}\right) \right\rfloor + 1. \tag{19}$$

**Remark:** Theorem 4 shows that SAder attains an $O(\sqrt{T(1+P_T)})$ bound for dynamic regret with switching cost, which is on the same order as that of Ader for dynamic regret. From the $\Omega(\sqrt{T(1+P_T)})$ lower bound of dynamic regret [Zhang et al., 2018a, Theorem 2], we know that our upper bound is optimal up to constant factors. Compared with the regret bound of OBD in (5) [Chen et al., 2018], the advantage of SAder is that its regret depends on the path-length $P_T$ directly, and thus becomes tighter when focusing on comparator sequences with smaller path-lengths. Finally, note that in (18), we did not minus the switching cost of the comparator sequence, i.e., $P_T$, that is because it is always smaller than $\sqrt{DT(1+P_T)}$ and does not affect the order.

---

**Algorithm 3** Lookahead SAder: Meta-algorithm

---

**Require:** A step size $\beta$, and a set $\mathcal{H}$ containing step sizes for experts
 1: Activate a set of experts $\{E^\eta | \eta \in \mathcal{H}\}$ by invoking the expert-algorithm for each step size $\eta \in \mathcal{H}$
 2: Sort step sizes in ascending order $\eta_1 \leq \eta_2 \leq \cdots \leq \eta_N$, and set $w_0^{\eta_i} = \frac{C}{i(i+1)}$
 3: **for** $t = 1, \ldots, T$ **do**
 4:     Observe the loss function $f_t(\cdot)$ and send it to each expert $E^\eta$
 5:     Receive $\mathbf{x}_t^\eta$ from each expert $E^\eta$
 6:     Update the weight of each expert by (20)
 7:     Output the weighted average $\mathbf{x}_t = \sum_{\eta \in \mathcal{H}} w_t^\eta \mathbf{x}_t^\eta$
 8: **end for**

---

**Algorithm 4** Lookahead SAder: Expert-algorithm

---

**Require:** The step size $\eta$
 1: **for** $t = 1, \ldots, T$ **do**
 2:     Receive the loss $f_t(\cdot)$ from the meta-algorithm
 3:     Solve the optimization problem in (21) to obtain $\mathbf{x}_t^\eta$
 4:     Submit $\mathbf{x}_t^\eta$ to the meta-algorithm
 5: **end for**

---

## 4.2 The lookahead setting

It is interesting to investigate whether we can do better if the hitting cost is available before predictions. In this case, we propose a *lookahead* version of SAder, and demonstrate that the regret bound remains on the same order, but Assumption 2 can be dropped. That is, the gradient of the function could be unbounded, and thus the function could also be unbounded.

**Meta-algorithm**   We design a lookahead version of Hedge, and summarize it in Algorithm 3. Compared with Algorithm 1, we make the following modifications.

- In the $t$-th round, the meta-algorithm first sends $f_t(\cdot)$ to all experts so that they can also benefit from the prior knowledge of $f_t(\cdot)$ (Step 4).
- After receiving the prediction from experts (Step 5), the meta-algorithm makes use of $f_t(\cdot)$ to determine the weights of experts (Step 6):

$$w_t^\eta = \frac{w_{t-1}^\eta e^{-\beta \ell_t(\mathbf{x}_t^\eta)}}{\sum_{\eta \in \mathcal{H}} w_{t-1}^\eta e^{-\beta \ell_t(\mathbf{x}_t^\eta)}} \tag{20}$$

where $\ell_t(\mathbf{x}_t^\eta)$ is defined in (15).

**Expert-algorithm**   To exploit the hitting cost of the current round, we choose an instance of the greedy algorithm in (4) as the expert-algorithm, and summarize it in Algorithm 4. The input of the expert is its step size $\eta$. After receiving $f_t(\cdot)$ (Step 2), the expert solves the following optimization problem to obtain $\mathbf{x}_t^\eta$ (Step 3):

$$\min_{\mathbf{x} \in \mathcal{X}} \quad f_t(\mathbf{x}) + \frac{1}{2\eta} \|\mathbf{x} - \mathbf{x}_{t-1}^\eta\|^2. \tag{21}$$

We have the following theoretical guarantee of the lookahead SAder.

**Theorem 5** *Set*

$$\mathcal{H} = \left\{ \eta_i = 2^{i-1} \sqrt{\frac{D^2}{T}} \,\middle|\, i = 1, \ldots, N \right\} \tag{22}$$

*where*

$$N = \left\lceil \frac{1}{2} \log_2(1 + 2T) \right\rceil + 1, \text{ and } \beta = \frac{1}{D}\sqrt{\frac{2}{T}}$$

*in Algorithm 3. Under Assumptions 1 and 3, for* any *comparator sequence* $\mathbf{u}_0, \mathbf{u}_1, \ldots, \mathbf{u}_T \in \mathcal{X}$, *the lookahead SAder satisfies*

$$\sum_{t=1}^{T} \Big( f_t(\mathbf{x}_t) + \|\mathbf{x}_t - \mathbf{x}_{t-1}\| \Big) - \sum_{t=1}^{T} f_t(\mathbf{u}_t)$$

$$\leq \frac{3}{2} \sqrt{T(D^2 + 2D \sum_{t=1}^{T} \|\mathbf{u}_t - \mathbf{u}_{t-1}\|) + D\sqrt{\frac{T}{2}} [1 + 2\ln(k+1)]}$$

$$= O\left( \sqrt{T(1 + P_T)} + \sqrt{T}(1 + \log\log P_T) \right) = O\left( \sqrt{T(1 + P_T)} \right)$$

*where* $\mathbf{x}_0 = 0$, *and* $k$ *is defined in (19).*

**Remark:** Similar to SAder, the lookahead SAder also achieves an $O(\sqrt{T(1 + P_T)})$ bound for dynamic regret with switching cost. In the lookahead setting, we do not need Assumption 2 any more, and the constants in Theorem 5 are independent from $G$.

**Lower Bound** To show the optimality of Theorem 5, we provide the lower bound of dynamic regret with switching cost under the lookahead setting.

**Theorem 6** *For any online algorithm with lookahead ability and any* $\tau \in [0, TD]$, *there exists a sequence of functions* $f_1, \ldots, f_T$ *and a sequence of comparators* $\mathbf{u}_1, \ldots, \mathbf{u}_T$ *satisfying Assumptions 1 and 3 such that (i) the path-length of* $\mathbf{u}_1, \ldots, \mathbf{u}_T$ *is at most* $\tau$ *and (ii) the dynamic regret with switching cost w.r.t.* $\mathbf{u}_1, \ldots, \mathbf{u}_T$ *is at least* $\Omega(\sqrt{T(D^2 + D\tau)})$.

**Remark:** The above theorem indicates an $\Omega(\sqrt{T(1 + P_T)})$ lower bound, which implies that the lookahead SAder is optimal up to constant factors. Thus, even in the lookahead setting, it is impossible to improve the $O(\sqrt{T(1 + P_T)})$ upper bound.

## 5 Conclusion and future work

We investigate the problem of smoothed online learning (SOL), and derive constant competitive ratio or sublinear dynamic regret with switching cost. For competitive ratio, we demonstrate that the naive approach, which only minimizes the hitting cost, is $\max(1, \frac{2}{\alpha})$-competitive for $\alpha$-polyhedral functions and $1 + \frac{4}{\lambda}$-competitive for $\lambda$-quadratic growth functions. Furthermore, we show that the greedy algorithm, which minimizes the weighted sum of the hitting cost and the switching cost, is $1 + \frac{2}{\sqrt{\lambda}}$-competitive for convex and $\lambda$-quadratic growth functions. For dynamic regret with switching cost, we propose smoothed Ader (SAder), which attains the optimal $O(\sqrt{T(1 + P_T)})$ bound. We also develop a lookahead version of SAder to make use of the prior knowledge of the hitting cost, and establish an $\Omega(\sqrt{T(1 + P_T)})$ lower bound.

The research on SOL is still on its early stage, and there are many open problems.

1. Although we can upper bound the sum of the hitting cost and the switching cost, we do not have a direct control over the switching cost. However, in many real problems, there may exist a hard constraint on the switching cost, motivating the study of switch-constrained online learning, in which the times of switches are limited [Altschuler and Talwar, 2018, Chen et al., 2020]. It would be interesting to investigate how to impose a budget on the switching cost [Wang et al., 2021].

2. This work investigates competitive ratio and dynamic regret with switching cost separately. To bound the two metrics simultaneously, one possible way is to create two experts which are designed for competitive ratio and dynamic regret with switching cost respectively, and then aggregate their predictions by the meta-algorithm of Daniely and Mansour [2019, Algorithm 2]. But we need to assume the hitting cost is bounded, because that meta-algorithm cannot make use of the lookahead ability.

3. As aforementioned, for polyhedral functions and quadratic growth functions, the best competitive ratio is obtained by the naive approach which ignores the switching cost, and this fact is counterintuitive. To better understand the challenge, it is important to reveal the lower bound of polyhedral functions and quadratic growth functions.

## Acknowledgments and Disclosure of Funding

The authors would like to thank Yuxuan Xiang for discussions about Theorem 4. Funding in direct support of this work: NSFC grant 62122037 and 61921006, JiangsuSF grant BK20200064.

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
