# Supplementary Material of Revisiting Smoothed Online Learning

**Lijun Zhang**[1,2], **Wei Jiang**[1], **Shiyin Lu**[1], **Tianbao Yang**[3]
[1]National Key Laboratory for Novel Software Technology, Nanjing University, Nanjing, China
[2]Peng Cheng Laboratory, Shenzhen, Guangdong, China
[3]Department of Computer Science, The University of Iowa, Iowa City, IA 52242, USA
{zhanglj, jiangw, lusy}@lamda.nju.edu.cn, tianbao-yang@uiowa.edu

## A  Analysis

In this section, we present the analysis of all the theorems.

### A.1  Proof of Theorem 1

Recall that $\mathbf{x}_t$ is the minimizer of $f_t(\cdot)$, which is $\alpha$-polyhedral. When $t \geq 2$, we have

$$
\begin{aligned}
&f_t(\mathbf{x}_t) + \|\mathbf{x}_t - \mathbf{x}_{t-1}\| \\
\leq& f_t(\mathbf{x}_t) + \|\mathbf{x}_t - \mathbf{u}_t\| + \|\mathbf{u}_t - \mathbf{u}_{t-1}\| + \|\mathbf{u}_{t-1} - \mathbf{x}_{t-1}\| \\
\overset{(7)}{\leq}& f_t(\mathbf{x}_t) + \frac{1}{\alpha}\big(f_t(\mathbf{u}_t) - f_t(\mathbf{x}_t)\big) + \frac{1}{\alpha}\big(f_{t-1}(\mathbf{u}_{t-1}) - f_{t-1}(\mathbf{x}_{t-1})\big) + \|\mathbf{u}_t - \mathbf{u}_{t-1}\|.
\end{aligned}
$$

For $t = 1$, we have

$$
\begin{aligned}
&f_1(\mathbf{x}_1) + \|\mathbf{x}_1 - \mathbf{x}_0\| \\
\leq& f_1(\mathbf{x}_1) + \|\mathbf{x}_1 - \mathbf{u}_1\| + \|\mathbf{u}_1 - \mathbf{u}_0\| + \|\mathbf{u}_0 - \mathbf{x}_0\| \\
=& f_1(\mathbf{x}_1) + \|\mathbf{x}_t - \mathbf{u}_1\| + \|\mathbf{u}_1 - \mathbf{u}_0\| \\
\overset{(7)}{\leq}& f_1(\mathbf{x}_1) + \frac{1}{\alpha}\big(f_1(\mathbf{u}_1) - f_1(\mathbf{x}_1)\big) + \|\mathbf{u}_1 - \mathbf{u}_0\|.
\end{aligned}
$$

Summing over all the iterations, we have

$$
\begin{aligned}
&\sum_{t=1}^{T} \big(f_t(\mathbf{x}_t) + \|\mathbf{x}_t - \mathbf{x}_{t-1}\|\big) \\
\leq& \sum_{t=1}^{T} f_t(\mathbf{x}_t) + \frac{1}{\alpha}\sum_{t=1}^{T}\big(f_t(\mathbf{u}_t) - f_t(\mathbf{x}_t)\big) + \frac{1}{\alpha}\sum_{t=2}^{T}\big(f_{t-1}(\mathbf{u}_{t-1}) - f_{t-1}(\mathbf{x}_{t-1})\big) + \sum_{t=1}^{T}\|\mathbf{u}_t - \mathbf{u}_{t-1}\| \\
\leq& \sum_{t=1}^{T} f_t(\mathbf{x}_t) + \frac{2}{\alpha}\sum_{t=1}^{T}\big(f_t(\mathbf{u}_t) - f_t(\mathbf{x}_t)\big) + \sum_{t=1}^{T}\|\mathbf{u}_t - \mathbf{u}_{t-1}\| \\
=& \frac{2}{\alpha}\sum_{t=1}^{T} f_t(\mathbf{u}_t) + \sum_{t=1}^{T}\|\mathbf{u}_t - \mathbf{u}_{t-1}\| + \sum_{t=1}^{T}\left(1 - \frac{2}{\alpha}\right)f_t(\mathbf{x}_t).
\end{aligned}
\tag{23}
$$

where the second inequality follows from the fact that $f_T(\mathbf{x}_T) \leq f_T(\mathbf{u}_T)$.

35th Conference on Neural Information Processing Systems (NeurIPS 2021).

Thus, if $\alpha \geq 2$, we have

$$\sum_{t=1}^{T} \big(f_t(\mathbf{x}_t) + \|\mathbf{x}_t - \mathbf{x}_{t-1}\|\big)$$

$$\overset{(8),(23)}{\leq} \frac{2}{\alpha} \sum_{t=1}^{T} f_t(\mathbf{u}_t) + \sum_{t=1}^{T} \|\mathbf{u}_t - \mathbf{u}_{t-1}\| + \sum_{t=1}^{T} \left(1 - \frac{2}{\alpha}\right) f_t(\mathbf{u}_t) \tag{24}$$

$$\leq \sum_{t=1}^{T} \big(f_t(\mathbf{u}_t) + \|\mathbf{u}_t - \mathbf{u}_{t-1}\|\big)$$

which implies the naive algorithm is 1-competitive. Otherwise, we have

$$\sum_{t=1}^{T} \big(f_t(\mathbf{x}_t) + \|\mathbf{x}_t - \mathbf{x}_{t-1}\|\big)$$

$$\overset{(23)}{\leq} \frac{2}{\alpha} \sum_{t=1}^{T} f_t(\mathbf{u}_t) + \sum_{t=1}^{T} \|\mathbf{u}_t - \mathbf{u}_{t-1}\| \leq \frac{2}{\alpha} \sum_{t=1}^{T} \big(f_t(\mathbf{u}_t) + \|\mathbf{u}_t - \mathbf{u}_{t-1}\|\big). \tag{25}$$

We complete the proof by combining (24) and (25).

## A.2 Proof of Theorem 2

We will make use of the following basic inequality of squared $\ell_2$-norm [Goel et al., 2019, Lemma 12].

$$\|\mathbf{x} + \mathbf{y}\|^2 \leq (1 + \rho)\|\mathbf{x}\|^2 + \left(1 + \frac{1}{\rho}\right)\|\mathbf{y}\|^2, \ \forall \rho > 0. \tag{26}$$

When $t \geq 2$, we have

$$f_t(\mathbf{x}_t) + \frac{1}{2}\|\mathbf{x}_t - \mathbf{x}_{t-1}\|^2$$

$$\overset{(26)}{\leq} f_t(\mathbf{x}_t) + \frac{1 + \rho}{2}\|\mathbf{u}_t - \mathbf{u}_{t-1}\|^2 + \frac{1}{2}\left(1 + \frac{1}{\rho}\right)\|\mathbf{x}_t - \mathbf{x}_{t-1} - \mathbf{u}_t + \mathbf{u}_{t-1}\|^2$$

$$\overset{(26)}{\leq} f_t(\mathbf{x}_t) + \frac{1 + \rho}{2}\|\mathbf{u}_t - \mathbf{u}_{t-1}\|^2 + \left(1 + \frac{1}{\rho}\right)\left(\|\mathbf{u}_t - \mathbf{x}_t\|^2 + \|\mathbf{u}_{t-1} - \mathbf{x}_{t-1}\|^2\right)$$

$$\overset{(9)}{\leq} f_t(\mathbf{x}_t) + \frac{1 + \rho}{2}\|\mathbf{u}_t - \mathbf{u}_{t-1}\|^2 + \frac{2}{\lambda}\left(1 + \frac{1}{\rho}\right)\left(f_t(\mathbf{u}_t) - f_t(\mathbf{x}_t) + f_{t-1}(\mathbf{u}_{t-1}) - f_{t-1}(\mathbf{x}_{t-1})\right).$$

For $t = 1$, we have

$$f_1(\mathbf{x}_1) + \frac{1}{2}\|\mathbf{x}_1 - \mathbf{x}_0\|^2 \overset{(26),(9)}{\leq} f_1(\mathbf{x}_1) + \frac{1 + \rho}{2}\|\mathbf{u}_1 - \mathbf{u}_0\|^2 + \frac{2}{\lambda}\left(1 + \frac{1}{\rho}\right)\left(f_1(\mathbf{u}_1) - f_1(\mathbf{x}_1)\right).$$

Summing over all the iterations, we have

$$\sum_{t=1}^{T} \left(f_t(\mathbf{x}_t) + \frac{1}{2}\|\mathbf{x}_t - \mathbf{x}_{t-1}\|^2\right)$$

$$\leq \sum_{t=1}^{T} f_t(\mathbf{x}_t) + \frac{1 + \rho}{2} \sum_{t=1}^{T} \|\mathbf{u}_t - \mathbf{u}_{t-1}\|^2 + \frac{2}{\lambda}\left(1 + \frac{1}{\rho}\right) \sum_{t=1}^{T} \left(f_t(\mathbf{u}_t) - f_t(\mathbf{x}_t)\right)$$

$$+ \frac{2}{\lambda}\left(1 + \frac{1}{\rho}\right) \sum_{t=2}^{T} \left(f_{t-1}(\mathbf{u}_{t-1}) - f_{t-1}(\mathbf{x}_{t-1})\right) \tag{27}$$

$$\leq \sum_{t=1}^{T} f_t(\mathbf{x}_t) + \frac{1 + \rho}{2} \sum_{t=1}^{T} \|\mathbf{u}_t - \mathbf{u}_{t-1}\|^2 + \frac{4}{\lambda}\left(1 + \frac{1}{\rho}\right) \sum_{t=1}^{T} \left(f_t(\mathbf{u}_t) - f_t(\mathbf{x}_t)\right)$$

$$= \frac{4}{\lambda}\left(1 + \frac{1}{\rho}\right) \sum_{t=1}^{T} f_t(\mathbf{u}_t) + \frac{1 + \rho}{2} \sum_{t=1}^{T} \|\mathbf{u}_t - \mathbf{u}_{t-1}\|^2 + \left(1 - \frac{4}{\lambda}\left(1 + \frac{1}{\rho}\right)\right) \sum_{t=1}^{T} f_t(\mathbf{x}_t).$$

First, we consider the case that

$$1 - \frac{4}{\lambda}\left(1 + \frac{1}{\rho}\right) \leq 0 \Leftrightarrow \frac{\lambda}{4} \leq 1 + \frac{1}{\rho} \tag{28}$$

and have

$$\sum_{t=1}^{T}\left(f_t(\mathbf{x}_t) + \frac{1}{2}\|\mathbf{x}_t - \mathbf{x}_{t-1}\|^2\right)$$

$$\overset{(27),(28)}{\leq} \frac{4}{\lambda}\left(1 + \frac{1}{\rho}\right)\sum_{t=1}^{T}f_t(\mathbf{u}_t) + \frac{1+\rho}{2}\sum_{t=1}^{T}\|\mathbf{u}_t - \mathbf{u}_{t-1}\|^2$$

$$\leq \max\left(\frac{4}{\lambda}\left(1 + \frac{1}{\rho}\right), 1 + \rho\right)\sum_{t=1}^{T}\left(f_t(\mathbf{u}_t) + \frac{1}{2}\|\mathbf{u}_t - \mathbf{u}_{t-1}\|^2\right).$$

To minimize the competitive ratio, we set

$$\frac{4}{\lambda}\left(1 + \frac{1}{\rho}\right) = 1 + \rho \Rightarrow \rho = \frac{4}{\lambda}$$

and obtain

$$\sum_{t=1}^{T}\left(f_t(\mathbf{x}_t) + \frac{1}{2}\|\mathbf{x}_t - \mathbf{x}_{t-1}\|^2\right) \leq \left(1 + \frac{4}{\lambda}\right)\sum_{t=1}^{T}\left(f_t(\mathbf{u}_t) + \frac{1}{2}\|\mathbf{u}_t - \mathbf{u}_{t-1}\|^2\right). \tag{29}$$

Next, we study the case that

$$1 - \frac{4}{\lambda}\left(1 + \frac{1}{\rho}\right) \geq 0 \Leftrightarrow \frac{\lambda}{4} \geq 1 + \frac{1}{\rho}$$

which only happens when $\lambda > 4$. Then, we have

$$\sum_{t=1}^{T}\left(f_t(\mathbf{x}_t) + \frac{1}{2}\|\mathbf{x}_t - \mathbf{x}_{t-1}\|^2\right) \overset{(8),(27)}{\leq} \sum_{t=1}^{T}f_t(\mathbf{u}_t) + \frac{1+\rho}{2}\sum_{t=1}^{T}\|\mathbf{u}_t - \mathbf{u}_{t-1}\|^2.$$

To minimize the competitive ratio, we set $\rho = \frac{4}{\lambda-4}$, and obtain

$$\sum_{t=1}^{T}\left(f_t(\mathbf{x}_t) + \frac{1}{2}\|\mathbf{x}_t - \mathbf{x}_{t-1}\|^2\right) \leq \frac{\lambda}{\lambda - 4}\sum_{t=1}^{T}\left(f_t(\mathbf{u}_t) + \frac{1}{2}\|\mathbf{u}_t - \mathbf{u}_{t-1}\|^2\right)$$

which is worse than (29). So, we keep (29) as the final result.

### A.3  Proof of Theorem 3

Since $f_t(\cdot)$ is convex, the objective function of (10) is $\gamma$-strongly convex. From the quadratic growth property of strongly convex functions [Hazan and Kale, 2011], we have

$$f_t(\mathbf{x}_t) + \frac{\gamma}{2}\|\mathbf{x}_t - \mathbf{x}_{t-1}\|^2 + \frac{\gamma}{2}\|\mathbf{u} - \mathbf{x}_t\|^2 \leq f_t(\mathbf{u}) + \frac{\gamma}{2}\|\mathbf{u} - \mathbf{x}_{t-1}\|^2, \ \forall \mathbf{u} \in \mathcal{X}. \tag{30}$$

Similar to previous studies [Bansal et al., 2015], the analysis uses an amortized local competitiveness argument, using the potential function $c\|\mathbf{x}_t - \mathbf{u}_t\|^2$. We proceed to bound $f_t(\mathbf{x}_t) + \frac{1}{2}\|\mathbf{x}_t - \mathbf{x}_{t-1}\|^2 + c\|\mathbf{x}_t - \mathbf{u}_t\|^2 - c\|\mathbf{x}_{t-1} - \mathbf{u}_{t-1}\|^2$, and have

$$f_t(\mathbf{x}_t) + \frac{1}{2}\|\mathbf{x}_t - \mathbf{x}_{t-1}\|^2 + c\|\mathbf{x}_t - \mathbf{u}_t\|^2 - c\|\mathbf{x}_{t-1} - \mathbf{u}_{t-1}\|^2$$

$$\overset{(26)}{\leq} f_t(\mathbf{x}_t) + \frac{1}{2}\|\mathbf{x}_t - \mathbf{x}_{t-1}\|^2 + c\left(2\|\mathbf{x}_t - \mathbf{v}_t\|^2 + 2\|\mathbf{v}_t - \mathbf{u}_t\|^2\right) - c\|\mathbf{x}_{t-1} - \mathbf{u}_{t-1}\|^2$$

$$\overset{(9)}{\leq} \left(1 + \frac{4c}{\lambda}\right)f_t(\mathbf{x}_t) + \frac{1}{2}\|\mathbf{x}_t - \mathbf{x}_{t-1}\|^2 + \frac{4c}{\lambda}f_t(\mathbf{u}_t) - c\|\mathbf{x}_{t-1} - \mathbf{u}_{t-1}\|^2$$

$$= \left(1 + \frac{4c}{\lambda}\right)\left(f_t(\mathbf{x}_t) + \frac{\lambda}{2(\lambda + 4c)}\|\mathbf{x}_t - \mathbf{x}_{t-1}\|^2\right) + \frac{4c}{\lambda}f_t(\mathbf{u}_t) - c\|\mathbf{x}_{t-1} - \mathbf{u}_{t-1}\|^2.$$

Suppose

$$\frac{\lambda}{\lambda + 4c} \leq \gamma, \tag{31}$$

we have

$$f_t(\mathbf{x}_t) + \frac{1}{2}\|\mathbf{x}_t - \mathbf{x}_{t-1}\|^2 + c\|\mathbf{x}_t - \mathbf{u}_t\|^2 - c\|\mathbf{x}_{t-1} - \mathbf{u}_{t-1}\|^2$$

$$\leq \left(1 + \frac{4c}{\lambda}\right)\left(f_t(\mathbf{x}_t) + \frac{\gamma}{2}\|\mathbf{x}_t - \mathbf{x}_{t-1}\|^2\right) + \frac{4c}{\lambda}f_t(\mathbf{u}_t) - c\|\mathbf{x}_{t-1} - \mathbf{u}_{t-1}\|^2$$

$$\overset{(30)}{\leq} \left(1 + \frac{4c}{\lambda}\right)\left(f_t(\mathbf{u}_t) + \frac{\gamma}{2}\|\mathbf{u}_t - \mathbf{x}_{t-1}\|^2 - \frac{\gamma}{2}\|\mathbf{u}_t - \mathbf{x}_t\|^2\right) + \frac{4c}{\lambda}f_t(\mathbf{u}_t) - c\|\mathbf{x}_{t-1} - \mathbf{u}_{t-1}\|^2$$

$$= \left(1 + \frac{8c}{\lambda}\right)f_t(\mathbf{u}_t) + \frac{\gamma(\lambda + 4c)}{2\lambda}\|\mathbf{u}_t - \mathbf{x}_{t-1}\|^2 - \frac{\gamma(\lambda + 4c)}{2\lambda}\|\mathbf{u}_t - \mathbf{x}_t\|^2 - c\|\mathbf{x}_{t-1} - \mathbf{u}_{t-1}\|^2.$$

Summing over all the iterations and assuming $\mathbf{x}_0 = \mathbf{u}_0$, we have

$$\sum_{t=1}^{T}\left(f_t(\mathbf{x}_t) + \frac{1}{2}\|\mathbf{x}_t - \mathbf{x}_{t-1}\|^2\right) + c\|\mathbf{x}_T - \mathbf{u}_T\|^2$$

$$\leq \left(1 + \frac{8c}{\lambda}\right)\sum_{t=1}^{T}f_t(\mathbf{u}_t) + \frac{\gamma(\lambda + 4c)}{2\lambda}\sum_{t=1}^{T}\|\mathbf{u}_t - \mathbf{x}_{t-1}\|^2$$

$$- \frac{\gamma(\lambda + 4c)}{2\lambda}\sum_{t=1}^{T}\|\mathbf{u}_t - \mathbf{x}_t\|^2 - c\sum_{t=1}^{T}\|\mathbf{x}_{t-1} - \mathbf{u}_{t-1}\|^2$$

$$\leq \left(1 + \frac{8c}{\lambda}\right)\sum_{t=1}^{T}f_t(\mathbf{u}_t) + \frac{\gamma(\lambda + 4c)}{2\lambda}\sum_{t=1}^{T}\|\mathbf{u}_t - \mathbf{x}_{t-1}\|^2 - \left(\frac{\gamma(\lambda + 4c)}{2\lambda} + c\right)\sum_{t=1}^{T}\|\mathbf{x}_{t-1} - \mathbf{u}_{t-1}\|^2$$

$$\overset{(26)}{\leq} \left(1 + \frac{8c}{\lambda}\right)\sum_{t=1}^{T}f_t(\mathbf{u}_t) + \frac{\gamma(\lambda + 4c)}{2\lambda}\sum_{t=1}^{T}\|\mathbf{u}_t - \mathbf{x}_{t-1}\|^2$$

$$- \left(\frac{\gamma(\lambda + 4c)}{2\lambda} + c\right)\sum_{t=1}^{T}\left(\frac{1}{1+\rho}\|\mathbf{x}_{t-1} - \mathbf{u}_t\|^2 - \frac{1}{\rho}\|\mathbf{u}_t - \mathbf{u}_{t-1}\|^2\right)$$

$$\leq \left(1 + \frac{8c}{\lambda}\right)\sum_{t=1}^{T}f_t(\mathbf{u}_t) + \left(\frac{\gamma(\lambda + 4c)}{2\lambda} + c\right)\frac{1}{\rho}\sum_{t=1}^{T}\|\mathbf{u}_t - \mathbf{u}_{t-1}\|^2$$

$$\leq \max\left(1 + \frac{8c}{\lambda}, \left(\frac{\gamma(\lambda + 4c)}{2\lambda} + c\right)\frac{2}{\rho}\right)\sum_{t=1}^{T}\left(f_t(\mathbf{u}_t) + \frac{1}{2}\|\mathbf{u}_t - \mathbf{u}_{t-1}\|^2\right)$$

where in the penultimate inequality we assume

$$\frac{\gamma(\lambda + 4c)}{2\lambda} \leq \left(\frac{\gamma(\lambda + 4c)}{2\lambda} + c\right)\frac{1}{1+\rho} \Leftrightarrow \frac{\gamma(\lambda + 4c)}{2\lambda} \leq \frac{c}{\rho}. \tag{32}$$

Next, we minimize the competitive ratio under the constraints in (31) and (32), which can be summarized as

$$\frac{\lambda}{\lambda + 4c} \leq \gamma \leq \frac{\lambda}{\lambda + 4c}\frac{2c}{\rho}.$$

We first set $c = \frac{\rho}{2}$ and $\gamma = \frac{\lambda}{\lambda+4c}$, and obtain

$$\sum_{t=1}^{T}\left(f_t(\mathbf{x}_t) + \frac{1}{2}\|\mathbf{x}_t - \mathbf{x}_{t-1}\|^2\right) \leq \max\left(1 + \frac{4\rho}{\lambda}, 1 + \frac{1}{\rho}\right)\sum_{t=1}^{T}\left(f_t(\mathbf{u}_t) + \frac{1}{2}\|\mathbf{u}_t - \mathbf{u}_{t-1}\|^2\right).$$

Then, we set

$$1 + \frac{4\rho}{\lambda} = 1 + \frac{1}{\rho} \Rightarrow \rho = \frac{\sqrt{\lambda}}{2}.$$

As a result, the competitive ratio is

$$1 + \frac{1}{\rho} = 1 + \frac{2}{\sqrt{\lambda}},$$

and the parameter is

$$\gamma = \frac{\lambda}{\lambda + 4c} = \frac{\lambda}{\lambda + 2\rho} = \frac{\lambda}{\lambda + \sqrt{\lambda}}.$$

### A.4 Proof of Theorem 4

The analysis is similar to the proof of Theorem 3 of Zhang et al. [2018a]. In the analysis, we need to specify the behavior of the meta-algorithm and expert-algorithm at $t = 0$. To simplify the presentation, we set

$$\mathbf{x}_0 = 0, \text{ and } \mathbf{x}_0^\eta = 0, \ \forall \eta \in \mathcal{H}. \tag{33}$$

First, we bound the dynamic regret with switching cost of the meta-algorithm w.r.t. all experts simultaneously.

**Lemma 1** *Under Assumptions 2 and 3, and setting $\beta = \frac{2}{(2G+1)D}\sqrt{\frac{2}{5T}}$, we have*

$$\sum_{t=1}^{T} \left( s_t(\mathbf{x}_t) + \|\mathbf{x}_t - \mathbf{x}_{t-1}\| \right) - \sum_{t=1}^{T} \left( s_t(\mathbf{x}_t^\eta) + \|\mathbf{x}_t^\eta - \mathbf{x}_{t-1}^\eta\| \right) \leq (2G+1)D\sqrt{\frac{5T}{8}} \left( \ln \frac{1}{w_1^\eta} + 1 \right) \tag{34}$$

*for each $\eta \in \mathcal{H}$.*

Next, we bound the dynamic regret with switching cost of each expert w.r.t. any comparator sequence $\mathbf{u}_0, \mathbf{u}_1, \ldots, \mathbf{u}_T \in \mathcal{X}$.

**Lemma 2** *Under Assumptions 2 and 3, we have*

$$\sum_{t=1}^{T} \left( s_t(\mathbf{x}_t^\eta) + \|\mathbf{x}_t^\eta - \mathbf{x}_{t-1}^\eta\| \right) - \sum_{t=1}^{T} s_t(\mathbf{u}_t) \leq \frac{D^2}{2\eta} + \frac{D}{\eta} \sum_{t=1}^{T} \|\mathbf{u}_t - \mathbf{u}_{t-1}\| + \eta T \left( \frac{G^2}{2} + G \right). \tag{35}$$

Then, we show that for any sequence of comparators $\mathbf{u}_0, \mathbf{u}_1, \ldots, \mathbf{u}_T \in \mathcal{X}$ there exists an $\eta_k \in \mathcal{H}$ such that the R.H.S. of (35) is almost minimal. If we minimize the R.H.S. of (35) exactly, the optimal step size is

$$\eta^*(P_T) = \sqrt{\frac{D^2 + 2DP_T}{T(G^2 + 2G)}}. \tag{36}$$

From Assumption 3, we have the following bound of the path-length

$$0 \leq P_T = \sum_{t=1}^{T} \|\mathbf{u}_t - \mathbf{u}_{t-1}\| \overset{(12)}{\leq} TD. \tag{37}$$

Thus

$$\sqrt{\frac{D^2}{T(G^2 + 2G)}} \leq \eta^*(P_T) \leq \sqrt{\frac{D^2 + 2TD^2}{T(G^2 + 2G)}}.$$

From our construction of $\mathcal{H}$ in (17), it is easy to verify that

$$\min \mathcal{H} = \sqrt{\frac{D^2}{T(G^2 + 2G)}}, \text{ and } \max \mathcal{H} \geq \sqrt{\frac{D^2 + 2TD^2}{T(G^2 + 2G)}}.$$

As a result, for any possible value of $P_T$, there exists a step size $\eta_k \in \mathcal{H}$ with $k$ defined in (19), such that

$$\eta_k = 2^{k-1}\sqrt{\frac{D^2}{T(G^2 + 2G)}} \leq \eta^*(P_T) \leq 2\eta_k. \tag{38}$$

Plugging $\eta_k$ into (35), the dynamic regret with switching cost of expert $E^{\eta_k}$ is given by

$$
\sum_{t=1}^{T} \left( s_t(\mathbf{x}_t^{\eta_k}) + \|\mathbf{x}_t^{\eta_k} - \mathbf{x}_{t-1}^{\eta_k}\| \right) - \sum_{t=1}^{T} s_t(\mathbf{u}_t)
$$

$$
\leq \frac{D^2}{2\eta_k} + \frac{D}{\eta_k} \sum_{t=1}^{T} \|\mathbf{u}_t - \mathbf{u}_{t-1}\| + \eta_k T \left( \frac{G^2}{2} + G \right)
\tag{39}
$$

$$
\overset{(38)}{\leq} \frac{D^2}{\eta^*(P_T)} + \frac{2D}{\eta^*(P_T)} \sum_{t=1}^{T} \|\mathbf{u}_t - \mathbf{u}_{t-1}\| + \eta^*(P_T) T \left( \frac{G^2}{2} + G \right)
$$

$$
\overset{(36)}{=} \frac{3}{2} \sqrt{T(G^2 + 2G)(D^2 + 2DP_T)}.
$$

From (13), we know the initial weight of expert $E^{\eta_k}$ is

$$
w_1^{\eta_k} = \frac{C}{k(k+1)} \geq \frac{1}{k(k+1)} \geq \frac{1}{(k+1)^2}.
$$

Combining with (34), we obtain the relative performance of the meta-algorithm w.r.t. expert $E^{\eta_k}$:

$$
\sum_{t=1}^{T} \left( s_t(\mathbf{x}_t) + \|\mathbf{x}_t - \mathbf{x}_{t-1}\| \right) - \sum_{t=1}^{T} \left( s_t(\mathbf{x}_t^{\eta_k}) + \|\mathbf{x}_t^{\eta_k} - \mathbf{x}_{t-1}^{\eta_k}\| \right) \leq (2G+1)D\sqrt{\frac{5T}{8}} \left[ 1 + 2\ln(k+1) \right].
\tag{40}
$$

From (39) and (40), we derive the following upper bound for dynamic regret with switching cost

$$
\sum_{t=1}^{T} \left( s_t(\mathbf{x}_t) + \|\mathbf{x}_t - \mathbf{x}_{t-1}\| \right) - \sum_{t=1}^{T} s_t(\mathbf{u}_t)
$$

$$
\leq \frac{3}{2} \sqrt{T(G^2 + 2G)(D^2 + 2DP_T)} + (2G+1)D\sqrt{\frac{5T}{8}} \left[ 1 + 2\ln(k+1) \right].
\tag{41}
$$

Finally, from Assumption 1, we have

$$
f_t(\mathbf{x}_t) - f_t(\mathbf{u}_t) \leq \langle \nabla f_t(\mathbf{x}_t), \mathbf{x}_t - \mathbf{u}_t \rangle \overset{(16)}{=} s_t(\mathbf{x}_t) - s_t(\mathbf{u}_t).
\tag{42}
$$

We complete the proof by combining (41) and (42).

### A.5    Proof of Theorem 5

The analysis is similar to that of Theorem 4. The difference is that we need to take into account the lookahead property of the meta-algorithm and the expert-algorithm.

First, we bound the dynamic regret with switching cost of the meta-algorithm w.r.t. all experts simultaneously.

**Lemma 3** *Under Assumption 3, and setting* $\beta = \frac{1}{D}\sqrt{\frac{2}{T}}$, *we have*

$$
\sum_{t=1}^{T} \left( s_t(\mathbf{x}_t) + \|\mathbf{x}_t - \mathbf{x}_{t-1}\| \right) - \sum_{t=1}^{T} \left( s_t(\mathbf{x}_t^{\eta}) + \|\mathbf{x}_t^{\eta} - \mathbf{x}_{t-1}^{\eta}\| \right) \leq D\sqrt{\frac{T}{2}} \left( \ln \frac{1}{w_0^{\eta}} + 1 \right)
\tag{43}
$$

*for each* $\eta \in \mathcal{H}$.

Combining Lemma 3 with Assumption 1, we have

$$
\sum_{t=1}^{T} \left( f_t(\mathbf{x}_t) + \|\mathbf{x}_t - \mathbf{x}_{t-1}\| \right) - \sum_{t=1}^{T} \left( f_t(\mathbf{x}_t^{\eta}) + \|\mathbf{x}_t^{\eta} - \mathbf{x}_{t-1}^{\eta}\| \right) \overset{(42),(43)}{\leq} D\sqrt{\frac{T}{2}} \left( \ln \frac{1}{w_0^{\eta}} + 1 \right)
\tag{44}
$$

for each $\eta \in \mathcal{H}$.

Next, we bound the dynamic regret with switching cost of each expert w.r.t. any comparator sequence $\mathbf{u}_0, \mathbf{u}_1, \ldots, \mathbf{u}_T \in \mathcal{X}$.

**Lemma 4** *Under Assumptions 1 and 3, we have*

$$\sum_{t=1}^{T} \left( f_t(\mathbf{x}_t^\eta) + \|\mathbf{x}_t^\eta - \mathbf{x}_{t-1}^\eta\| \right) - \sum_{t=1}^{T} f_t(\mathbf{u}_t) \leq \frac{D^2}{2\eta} + \frac{D}{\eta} \sum_{t=1}^{T} \|\mathbf{u}_t - \mathbf{u}_{t-1}\| + \frac{\eta T}{2}. \quad (45)$$

The rest of the proof is almost identical to that of Theorem 4. We will show that for any sequence of comparators $\mathbf{u}_0, \mathbf{u}_1, \ldots, \mathbf{u}_T \in \mathcal{X}$ there exists an $\eta_k \in \mathcal{H}$ such that the R.H.S. of (45) is almost minimal. If we minimize the R.H.S. of (45) exactly, the optimal step size is

$$\eta^*(P_T) = \sqrt{\frac{D^2 + 2DP_T}{T}}. \quad (46)$$

From (37), we know that

$$\sqrt{\frac{D^2}{T}} \leq \eta^*(P_T) \leq \sqrt{\frac{D^2 + 2TD^2}{T}}.$$

From our construction of $\mathcal{H}$ in (22), it is easy to verify that

$$\min \mathcal{H} = \sqrt{\frac{D^2}{T}}, \text{ and } \max \mathcal{H} \geq \sqrt{\frac{D^2 + 2TD^2}{T}}.$$

As a result, for any possible value of $P_T$, there exists a step size $\eta_k \in \mathcal{H}$ with $k$ defined in (19), such that

$$\eta_k = 2^{k-1}\sqrt{\frac{D^2}{T}} \leq \eta^*(P_T) \leq 2\eta_k. \quad (47)$$

Plugging $\eta_k$ into (45), the dynamic regret with switching cost of expert $E^{\eta_k}$ is given by

$$\begin{aligned}
&\sum_{t=1}^{T} \left( f_t(\mathbf{x}_t^{\eta_k}) + \|\mathbf{x}_t^{\eta_k} - \mathbf{x}_{t-1}^{\eta_k}\| \right) - \sum_{t=1}^{T} f_t(\mathbf{u}_t) \\
&\leq \frac{D^2}{2\eta_k} + \frac{D}{\eta_k} \sum_{t=1}^{T} \|\mathbf{u}_t - \mathbf{u}_{t-1}\| + \frac{\eta_k T}{2} \\
&\overset{(47)}{\leq} \frac{D^2}{\eta^*(P_T)} + \frac{2D}{\eta^*(P_T)} \sum_{t=1}^{T} \|\mathbf{u}_t - \mathbf{u}_{t-1}\| + \frac{\eta^*(P_T)T}{2} \\
&\overset{(46)}{=} \frac{3}{2} \sqrt{T(D^2 + 2DP_T)}.
\end{aligned} \quad (48)$$

From Step 2 of Algorithm 3, we know the initial weight of expert $E^{\eta_k}$ is

$$w_0^{\eta_k} = \frac{C}{k(k+1)} \geq \frac{1}{k(k+1)} \geq \frac{1}{(k+1)^2}.$$

Combining with (44), we obtain the relative performance of the meta-algorithm w.r.t. expert $E^{\eta_k}$:

$$\sum_{t=1}^{T} \left( f_t(\mathbf{x}_t) + \|\mathbf{x}_t - \mathbf{x}_{t-1}\| \right) - \sum_{t=1}^{T} \left( f_t(\mathbf{x}_t^{\eta_k}) + \|\mathbf{x}_t^{\eta_k} - \mathbf{x}_{t-1}^{\eta_k}\| \right) \leq D\sqrt{\frac{T}{2}} \left[ 1 + 2\ln(k+1) \right]. \quad (49)$$

We complete the proof by summing (48) and (49) together.

### A.6 Proof of Theorem 6

The proof is built upon a lower bound of competitive ratio [Argue et al., 2020a]. By setting $\gamma = \frac{D}{2\sqrt{d}}$ in Lemma 12 of Argue et al. [2020a], we can guarantee that Assumption 3 is satisfied. Then, we choose $\mu = 0$, $\lambda = 1/\gamma$ in that lemma, and obtain the conclusion below.

**Lemma 5** *For any online algorithm $A$ and any fixed value of $d$, there exists a sequence of convex functions $f_1(\cdot), \ldots, f_d(\cdot)$ over the domain $[-\frac{D}{2\sqrt{d}}, \frac{D}{2\sqrt{d}}]^d$ in the lookahead setting such that*

1. *the sum of the hitting cost and the switching cost of A is at least $\frac{3\gamma d}{4} = \frac{3D\sqrt{d}}{8}$;*
2. *there exist a fixed point $\mathbf{u}$ whose hitting cost is $0$.*

We consider two cases: $\tau < D$ and $\tau \geq D$. When $\tau < D$, from Lemma 5 with $d = T$, we know that the dynamic regret with switching cost w.r.t. a fixed point $\mathbf{u}$ is at least $\Omega(D\sqrt{T})$.

Next, we consider the case $\tau \geq D$. Without loss of generality, we assume $\lfloor \tau/D \rfloor$ divides $T$. Then, we partition $T$ into $\lfloor \tau/D \rfloor$ successive stages, each of which contains $T/\lfloor \tau/D \rfloor$ rounds. Applying Lemma 5 to each stage, we conclude that there exists a sequence of convex functions $f_1(\cdot), \ldots, f_T(\cdot)$ over the domain $[-\frac{D}{2\sqrt{d}}, \frac{D}{2\sqrt{d}}]^d$ where $d = T/\lfloor \tau/D \rfloor$ in the lookahead setting such that

1. the sum of the hitting cost and the switching cost of any online algorithm is at least

$$\lfloor \tau/D \rfloor \cdot \frac{3D}{8}\sqrt{\frac{T}{\lfloor \tau/D \rfloor}} = \frac{3D}{8}\sqrt{T\left\lfloor\frac{\tau}{D}\right\rfloor} = \Omega(\sqrt{TD\tau});$$

2. there exists a sequence of points $\mathbf{u}_1, \ldots, \mathbf{u}_T$ whose hitting cost is $0$ and switching cost (i.e., path-length) is at most

$$D\left\lfloor\frac{\tau}{D}\right\rfloor \leq \tau$$

since they switch at most $\lfloor \tau/D \rfloor - 1$ times.

Thus, the dynamic regret with switching cost w.r.t. $\mathbf{u}_1, \ldots, \mathbf{u}_T$ is at least

$$\frac{3D}{8}\sqrt{T\left\lfloor\frac{\tau}{D}\right\rfloor} - \tau = \Omega(\sqrt{TD\tau}).$$

We complete the proof by combining the results of the above two cases.

# B  Proof of supporting lemmas

We provide the proof of all the supporting lemmas.

## B.1  Proof of Lemma 1

Based on the prediction rule of the meta-algorithm, we upper bound the switching cost when $t \geq 2$ as follows:

$$
\begin{aligned}
\|\mathbf{x}_t - \mathbf{x}_{t-1}\| &= \left\|\sum_{\eta \in \mathcal{H}} w_t^\eta \mathbf{x}_t^\eta - \sum_{\eta \in \mathcal{H}} w_{t-1}^\eta \mathbf{x}_{t-1}^\eta\right\| = \left\|\sum_{\eta \in \mathcal{H}} w_t^\eta (\mathbf{x}_t^\eta - \mathbf{x}) - \sum_{\eta \in \mathcal{H}} w_{t-1}^\eta (\mathbf{x}_{t-1}^\eta - \mathbf{x})\right\| \\
&\leq \left\|\sum_{\eta \in \mathcal{H}} w_t^\eta (\mathbf{x}_t^\eta - \mathbf{x}) - \sum_{\eta \in \mathcal{H}} w_t^\eta (\mathbf{x}_{t-1}^\eta - \mathbf{x})\right\| + \left\|\sum_{\eta \in \mathcal{H}} w_t^\eta (\mathbf{x}_{t-1}^\eta - \mathbf{x}) - \sum_{\eta \in \mathcal{H}} w_{t-1}^\eta (\mathbf{x}_{t-1}^\eta - \mathbf{x})\right\| \\
&= \left\|\sum_{\eta \in \mathcal{H}} w_t^\eta (\mathbf{x}_t^\eta - \mathbf{x}_{t-1}^\eta)\right\| + \left\|\sum_{\eta \in \mathcal{H}} (w_t^\eta - w_{t-1}^\eta)(\mathbf{x}_{t-1}^\eta - \mathbf{x})\right\| \\
&\leq \sum_{\eta \in \mathcal{H}} w_t^\eta \left\|\mathbf{x}_t^\eta - \mathbf{x}_{t-1}^\eta\right\| + \sum_{\eta \in \mathcal{H}} |w_t^\eta - w_{t-1}^\eta| \left\|\mathbf{x}_{t-1}^\eta - \mathbf{x}\right\| \\
&\overset{(12)}{\leq} \sum_{\eta \in \mathcal{H}} w_t^\eta \left\|\mathbf{x}_t^\eta - \mathbf{x}_{t-1}^\eta\right\| + D \sum_{\eta \in \mathcal{H}} |w_t^\eta - w_{t-1}^\eta| = \sum_{\eta \in \mathcal{H}} w_t^\eta \left\|\mathbf{x}_t^\eta - \mathbf{x}_{t-1}^\eta\right\| + D\|\mathbf{w}_t - \mathbf{w}_{t-1}\|_1
\end{aligned}
$$

$$(50)$$

where $\mathbf{x}$ is an arbitrary point in $\mathcal{X}$, and $\mathbf{w}_t = (w_t^\eta)_{\eta \in \mathcal{H}} \in \mathbb{R}^N$. When $t = 1$, from (33), we have

$$\|\mathbf{x}_1 - \mathbf{x}_0\| = \|\mathbf{x}_1\| = \left\|\sum_{\eta \in \mathcal{H}} w_1^\eta \mathbf{x}_1^\eta\right\| \leq \sum_{\eta \in \mathcal{H}} w_1^\eta \|\mathbf{x}_1^\eta\| = \sum_{\eta \in \mathcal{H}} w_1^\eta \|\mathbf{x}_1^\eta - \mathbf{x}_0^\eta\|. \tag{51}$$

Then, the relative loss of the meta-algorithm w.r.t. expert $E^\eta$ can be decomposed as

$$\sum_{t=1}^{T}\Big(s_t(\mathbf{x}_t) + \|\mathbf{x}_t - \mathbf{x}_{t-1}\|\Big) - \sum_{t=1}^{T}\Big(s_t(\mathbf{x}_t^\eta) + \|\mathbf{x}_t^\eta - \mathbf{x}_{t-1}^\eta\|\Big)$$

$$\overset{(16),(50),(51)}{\leq} \sum_{t=1}^{T}\left(\sum_{\eta\in\mathcal{H}} w_t^\eta \|\mathbf{x}_t^\eta - \mathbf{x}_{t-1}^\eta\| - \Big(\langle\nabla f_t(\mathbf{x}_t), \mathbf{x}_t^\eta - \mathbf{x}_t\rangle + \|\mathbf{x}_t^\eta - \mathbf{x}_{t-1}^\eta\|\Big)\right)$$
$$+ D\sum_{t=2}^{T}\|\mathbf{w}_t - \mathbf{w}_{t-1}\|_1 \tag{52}$$

$$\overset{(15)}{=} \underbrace{\sum_{t=1}^{T}\left(\sum_{\eta\in\mathcal{H}} w_t^\eta \ell_t(\mathbf{x}_t^\eta) - \ell_t(\mathbf{x}_t^\eta)\right)}_{:=A} + D\sum_{t=2}^{T}\|\mathbf{w}_t - \mathbf{w}_{t-1}\|_1.$$

We proceed to bound $A$ and $\|\mathbf{w}_t - \mathbf{w}_{t-1}\|_1$ in (52). Notice that $A$ is the regret of the meta-algorithm w.r.t. expert $E^\eta$. From Assumptions 2 and 3, we have

$$|\langle\nabla f_t(\mathbf{x}_t), \mathbf{x}_t^\eta - \mathbf{x}_t\rangle| \leq \|\nabla f_t(\mathbf{x}_t)\|\|\mathbf{x}_t^\eta - \mathbf{x}_t\| \overset{(11),(12)}{\leq} GD.$$

Thus, we have

$$-GD \leq \ell_t(\mathbf{x}_t^\eta) \leq (G+1)D, \ \forall\eta\in\mathcal{H}. \tag{53}$$

According to the standard analysis of Hedge [Zhang et al., 2018a, Lemma 1] and (53), we have

$$\sum_{t=1}^{T}\left(\sum_{\eta\in\mathcal{H}} w_t^\eta \ell_t(\mathbf{x}_t^\eta) - \ell_t(\mathbf{x}_t^\eta)\right) \leq \frac{1}{\beta}\ln\frac{1}{w_1^\eta} + \frac{\beta T(2G+1)^2 D^2}{8}. \tag{54}$$

Next, we bound $\|\mathbf{w}_t - \mathbf{w}_{t-1}\|_1$, which measures the stability of the meta-algorithm, i.e., the change of coefficients between successive rounds. Because the Hedge algorithm is translation invariant, we can subtract $D/2$ from $\ell_t(\mathbf{x}_t^\eta)$ such that

$$|\ell_t(\mathbf{x}_t^\eta) - D/2| \leq (G+1/2)D, \ \forall\eta\in\mathcal{H}. \tag{55}$$

It is well-known that Hedge can be treated as a special case of "Follow-the-Regularized-Leader" with entropic regularization [Shalev-Shwartz, 2011]

$$R(\mathbf{w}) = \sum_i w_i \log w_i$$

over the probability simplex, and $R(\cdot)$ is 1-strongly convex w.r.t. the $\ell_1$-norm. In other words, we have

$$\mathbf{w}_{t+1} = \underset{\mathbf{w}\in\Delta}{\arg\min}\left\langle -\frac{1}{\beta}\log(\mathbf{w}_1) + \sum_{i=1}^{t}\mathbf{g}_i, \mathbf{w}\right\rangle + \frac{1}{\beta}R(\mathbf{w}), \ \forall t\geq 1$$

where $\Delta\subseteq\mathbb{R}^N$ is the probability simplex, and $\mathbf{g}_i = [\ell_i(\mathbf{x}_i^\eta) - D/2]_{\eta\in\mathcal{H}} \in \mathbb{R}^N$. From the stability property of Follow-the-Regularized-Leader [Duchi et al., 2012, Lemma 2], we have

$$\|\mathbf{w}_t - \mathbf{w}_{t-1}\|_1 \leq \beta\|\mathbf{g}_{t-1}\|_\infty \overset{(55)}{\leq} \beta(G+1/2)D, \ \forall t\geq 2.$$

Then

$$\sum_{t=2}^{T}\|\mathbf{w}_t - \mathbf{w}_{t-1}\|_1 \leq \frac{\beta(T-1)(2G+1)D}{2}. \tag{56}$$

Substituting (54) and (56) into (52), we have

$$\sum_{t=1}^{T}\Big(s_t(\mathbf{x}_t) + \|\mathbf{x}_t - \mathbf{x}_{t-1}\|\Big) - \sum_{t=1}^{T}\Big(s_t(\mathbf{x}_t^\eta) + \|\mathbf{x}_t^\eta - \mathbf{x}_{t-1}^\eta\|\Big)$$

$$\leq \frac{1}{\beta}\ln\frac{1}{w_1^\eta} + \frac{\beta T(2G+1)^2 D^2}{8} + \frac{\beta(T-1)(2G+1)D^2}{2} \leq \frac{1}{\beta}\ln\frac{1}{w_1^\eta} + \frac{5\beta T(2G+1)^2 D^2}{8}.$$

We complete the proof by setting $\beta = \frac{2}{(2G+1)D}\sqrt{\frac{2}{5T}}$.

## B.2 Proof of Lemma 2

First, we bound the dynamic regret of the expert-algorithm. Define
$$\bar{\mathbf{x}}_{t+1}^{\eta} = \mathbf{x}_t^{\eta} - \eta \nabla f_t(\mathbf{x}_t).$$
Following the analysis of Ader [Zhang et al., 2018a, Theorems 1 and 6], we have

$$s_t(\mathbf{x}_t^{\eta}) - s_t(\mathbf{u}_t) \overset{(16)}{=} \langle \nabla f_t(\mathbf{x}_t), \mathbf{x}_t^{\eta} - \mathbf{u}_t \rangle = \frac{1}{\eta} \langle \mathbf{x}_t^{\eta} - \bar{\mathbf{x}}_{t+1}^{\eta}, \mathbf{x}_t^{\eta} - \mathbf{u}_t \rangle$$

$$= \frac{1}{2\eta} \left( \|\mathbf{x}_t^{\eta} - \mathbf{u}_t\|_2^2 - \|\bar{\mathbf{x}}_{t+1}^{\eta} - \mathbf{u}_t\|_2^2 + \|\mathbf{x}_t^{\eta} - \bar{\mathbf{x}}_{t+1}^{\eta}\|_2^2 \right)$$

$$= \frac{1}{2\eta} \left( \|\mathbf{x}_t^{\eta} - \mathbf{u}_t\|_2^2 - \|\bar{\mathbf{x}}_{t+1}^{\eta} - \mathbf{u}_t\|_2^2 \right) + \frac{\eta}{2} \|\nabla f_t(\mathbf{x}_t)\|_2^2$$

$$\overset{(11)}{\leq} \frac{1}{2\eta} \left( \|\mathbf{x}_t^{\eta} - \mathbf{u}_t\|_2^2 - \|\bar{\mathbf{x}}_{t+1}^{\eta} - \mathbf{u}_t\|_2^2 \right) + \frac{\eta}{2} G^2$$

$$\leq \frac{1}{2\eta} \left( \|\mathbf{x}_t^{\eta} - \mathbf{u}_t\|_2^2 - \|\mathbf{x}_{t+1}^{\eta} - \mathbf{u}_t\|_2^2 \right) + \frac{\eta}{2} G^2$$

$$= \frac{1}{2\eta} \left( \|\mathbf{x}_t^{\eta} - \mathbf{u}_t\|_2^2 - \|\mathbf{x}_{t+1}^{\eta} - \mathbf{u}_{t+1}\|_2^2 + \|\mathbf{x}_{t+1}^{\eta} - \mathbf{u}_{t+1}\|_2^2 - \|\mathbf{x}_{t+1}^{\eta} - \mathbf{u}_t\|_2^2 \right) + \frac{\eta}{2} G^2$$

$$= \frac{1}{2\eta} \left( \|\mathbf{x}_t^{\eta} - \mathbf{u}_t\|_2^2 - \|\mathbf{x}_{t+1}^{\eta} - \mathbf{u}_{t+1}\|_2^2 + (\mathbf{x}_{t+1}^{\eta} - \mathbf{u}_{t+1} + \mathbf{x}_{t+1}^{\eta} - \mathbf{u}_t)^{\top} (\mathbf{u}_t - \mathbf{u}_{t+1}) \right) + \frac{\eta}{2} G^2$$

$$\leq \frac{1}{2\eta} \left( \|\mathbf{x}_t^{\eta} - \mathbf{u}_t\|_2^2 - \|\mathbf{x}_{t+1}^{\eta} - \mathbf{u}_{t+1}\|_2^2 + \left( \|\mathbf{x}_{t+1}^{\eta} - \mathbf{u}_{t+1}\| + \|\mathbf{x}_{t+1}^{\eta} - \mathbf{u}_t\| \right) \|\mathbf{u}_t - \mathbf{u}_{t+1}\| \right) + \frac{\eta}{2} G^2$$

$$\overset{(12)}{\leq} \frac{1}{2\eta} \left( \|\mathbf{x}_t^{\eta} - \mathbf{u}_t\|_2^2 - \|\mathbf{x}_{t+1}^{\eta} - \mathbf{u}_{t+1}\|_2^2 \right) + \frac{D}{\eta} \|\mathbf{u}_t - \mathbf{u}_{t+1}\| + \frac{\eta}{2} G^2.$$

Summing the above inequality over all iterations, we have

$$\sum_{t=1}^{T} \left( s_t(\mathbf{x}_t^{\eta}) - s_t(\mathbf{u}_t) \right) \leq \frac{1}{2\eta} \|\mathbf{x}_1^{\eta} - \mathbf{u}_1\|_2^2 + \frac{D}{\eta} \sum_{t=1}^{T} \|\mathbf{u}_{t+1} - \mathbf{u}_t\| + \frac{\eta T}{2} G^2$$

$$\overset{(12)}{\leq} \frac{1}{2\eta} D^2 + \frac{D}{\eta} \sum_{t=1}^{T} \|\mathbf{u}_{t+1} - \mathbf{u}_t\| + \frac{\eta T}{2} G^2. \tag{57}$$

Since (57) holds when $\mathbf{u}_{T+1} = \mathbf{u}_T$, we have

$$\sum_{t=1}^{T} \left( s_t(\mathbf{x}_t^{\eta}) - s_t(\mathbf{u}_t) \right) \leq \frac{1}{2\eta} D^2 + \frac{D}{\eta} \sum_{t=1}^{T} \|\mathbf{u}_t - \mathbf{u}_{t-1}\| + \frac{\eta T}{2} G^2. \tag{58}$$

Next, we bound the switching cost of the expert-algorithm. To this end, we have

$$\sum_{t=1}^{T} \|\mathbf{x}_t^{\eta} - \mathbf{x}_{t-1}^{\eta}\| = \sum_{t=0}^{T-1} \|\mathbf{x}_{t+1}^{\eta} - \mathbf{x}_t^{\eta}\| \leq \sum_{t=0}^{T-1} \|\bar{\mathbf{x}}_{t+1}^{\eta} - \mathbf{x}_t^{\eta}\| = \sum_{t=0}^{T-1} \|\eta \nabla f_t(\mathbf{x}_t)\| \overset{(11)}{\leq} \eta T G. \tag{59}$$

We complete the proof by combining (58) with (59).

## B.3 Proof of Lemma 3

We reuse the first part of the proof of Lemma 1, and start from (52). To bound $A$, we need to analyze the behavior of the lookahead Hedge. To this end, we prove the following lemma.

**Lemma 6** *The meta-algorithm in Algorithm 3 satisfies*

$$\sum_{t=1}^{T} \left( \sum_{\eta \in \mathcal{H}} w_t^{\eta} \ell_t(\mathbf{x}_t^{\eta}) - \ell_t(\mathbf{x}_t^{\eta}) \right) \leq \frac{1}{\beta} \ln \frac{1}{w_0^{\eta}} - \frac{1}{2\beta} \sum_{t=1}^{T} \|\mathbf{w}_t - \mathbf{w}_{t-1}\|_1^2 \tag{60}$$

*for any $\eta \in \mathcal{H}$.*

Substituting (60) into (52), we have

$$\sum_{t=1}^{T}\Big(s_t(\mathbf{x}_t)+\|\mathbf{x}_t-\mathbf{x}_{t-1}\|\Big)-\sum_{t=1}^{T}\Big(s_t(\mathbf{x}_t^{\eta})+\|\mathbf{x}_t^{\eta}-\mathbf{x}_{t-1}^{\eta}\|\Big)$$

$$\leq\frac{1}{\beta}\ln\frac{1}{w_0^{\eta}}-\frac{1}{2\beta}\sum_{t=1}^{T}\|\mathbf{w}_t-\mathbf{w}_{t-1}\|_1^2+D\sum_{t=2}^{T}\|\mathbf{w}_t-\mathbf{w}_{t-1}\|_1$$

$$\leq\frac{1}{\beta}\ln\frac{1}{w_0^{\eta}}-\frac{1}{2\beta}\sum_{t=1}^{T}\|\mathbf{w}_t-\mathbf{w}_{t-1}\|_1^2+\sum_{t=2}^{T}\left(\frac{1}{2\beta}\|\mathbf{w}_t-\mathbf{w}_{t-1}\|_1^2+\frac{\beta D^2}{2}\right)$$

$$\leq\frac{1}{\beta}\ln\frac{1}{w_0^{\eta}}+\frac{\beta T D^2}{2}=D\sqrt{\frac{T}{2}}\left(\ln\frac{1}{w_0^{\eta}}+1\right)$$

(61)

where we set $\beta=\frac{1}{D}\sqrt{\frac{2}{T}}$.

## B.4 Proof of Lemma 6

To simplify the notation, we define

$$W_0=\sum_{\eta\in\mathcal{H}}w_0^{\eta}=1,\; L_t^{\eta}=\sum_{i=1}^{t}\ell_i(\mathbf{x}_i^{\eta}),\; \text{and }W_t=\sum_{\eta\in\mathcal{H}}w_0^{\eta}e^{-\beta L_t^{\eta}},\;\forall t\geq 1.$$

From the updating rule in (20), it is easy to verify that

$$w_t^{\eta}=\frac{w_0^{\eta}e^{-\beta L_t^{\eta}}}{W_t},\;\forall t\geq 1.$$

(62)

First, we have

$$\ln W_T=\ln\left(\sum_{\eta\in\mathcal{H}}w_0^{\eta}e^{-\beta L_T^{\eta}}\right)\geq\ln\left(\max_{\eta\in\mathcal{H}}w_0^{\eta}e^{-\beta L_T^{\eta}}\right)=-\beta\min_{\eta\in\mathcal{H}}\left(L_T^{\eta}+\frac{1}{\beta}\ln\frac{1}{w_0^{\eta}}\right).$$

(63)

Next, we bound the related quantity $\ln(W_t/W_{t-1})$ as follows. For any $\eta\in\mathcal{H}$, we have

$$\ln\left(\frac{W_t}{W_{t-1}}\right)\overset{(62)}{=}\ln\left(\frac{w_0^{\eta}e^{-\beta L_t^{\eta}}}{w_t^{\eta}}\frac{w_{t-1}^{\eta}}{w_0^{\eta}e^{-\beta L_{t-1}^{\eta}}}\right)=\ln\left(\frac{w_{t-1}^{\eta}}{w_t^{\eta}}\right)-\beta\ell_t(\mathbf{x}_t^{\eta}).$$

(64)

Then, we have

$$\ln\left(\frac{W_t}{W_{t-1}}\right)=\ln\left(\frac{W_t}{W_{t-1}}\right)\sum_{\eta\in\mathcal{H}}w_t^{\eta}=\sum_{\eta\in\mathcal{H}}w_t^{\eta}\ln\left(\frac{W_t}{W_{t-1}}\right)$$

$$\overset{(64)}{=}\sum_{\eta\in\mathcal{H}}w_t^{\eta}\ln\left(\frac{w_{t-1}^{\eta}}{w_t^{\eta}}\right)-\beta\sum_{\eta\in\mathcal{H}}w_t^{\eta}\ell_t(\mathbf{x}_t^{\eta})\leq-\frac{1}{2}\|\mathbf{w}_t-\mathbf{w}_{t-1}\|_1^2-\beta\sum_{\eta\in\mathcal{H}}w_t^{\eta}\ell_t(\mathbf{x}_t^{\eta})$$

(65)

where the last inequality is due to Pinsker's inequality [Cover and Thomas, 2006, Lemma 11.6.1]. Thus

$$\ln W_T=\ln W_0+\sum_{t=1}^{T}\ln\left(\frac{W_t}{W_{t-1}}\right)\overset{(65)}{=}\sum_{t=1}^{T}\left(-\frac{1}{2}\|\mathbf{w}_t-\mathbf{w}_{t-1}\|_1^2-\beta\sum_{\eta\in\mathcal{H}}w_t^{\eta}\ell_t(\mathbf{x}_t^{\eta})\right).$$

(66)

Combining (63) with (66), we obtain

$$-\beta\min_{\eta\in\mathcal{H}}\left(L_T^{\eta}+\frac{1}{\beta}\ln\frac{1}{w_0^{\eta}}\right)\leq\sum_{t=1}^{T}\left(-\frac{1}{2}\|\mathbf{w}_t-\mathbf{w}_{t-1}\|_1^2-\beta\sum_{\eta\in\mathcal{H}}w_t^{\eta}\ell_t(\mathbf{x}_t^{\eta})\right)$$

We complete the proof by rearranging the above inequality.

## B.5 Proof of Lemma 4

The analysis is similar to that of Theorem 10 of Chen et al. [2018], which relies on a strong condition

$$\mathbf{x}_t^\eta = \mathbf{x}_{t-1}^\eta - \eta \nabla f_t(\mathbf{x}_t^\eta).$$

Note that the above equation is essentially the vanishing gradient condition of $\mathbf{x}_t^\eta$ when (21) is unconstrained. In contrast, we only make use of the first-order optimality criterion of $\mathbf{x}_t^\eta$ [Boyd and Vandenberghe, 2004], i.e.,

$$\left\langle \nabla f_t(\mathbf{x}_t^\eta) + \frac{1}{\eta}(\mathbf{x}_t^\eta - \mathbf{x}_{t-1}^\eta), \mathbf{y} - \mathbf{x}_t^\eta \right\rangle \geq 0, \ \forall \mathbf{y} \in \mathcal{X} \tag{67}$$

which is much weaker.

From the convexity of $f_t(\cdot)$, we have

$$f_t(\mathbf{x}_t^\eta) - f_t(\mathbf{u}_t)$$
$$\leq \langle \nabla f_t(\mathbf{x}_t^\eta), \mathbf{x}_t^\eta - \mathbf{u}_t \rangle$$
$$\overset{(67)}{\leq} \frac{1}{\eta}\langle \mathbf{x}_t^\eta - \mathbf{x}_{t-1}^\eta, \mathbf{u}_t - \mathbf{x}_t^\eta \rangle = \frac{1}{2\eta}\left( \|\mathbf{x}_{t-1}^\eta - \mathbf{u}_t\|^2 - \|\mathbf{x}_t^\eta - \mathbf{u}_t\|^2 - \|\mathbf{x}_t^\eta - \mathbf{x}_{t-1}^\eta\|^2 \right)$$
$$= \frac{1}{2\eta}\left( \|\mathbf{x}_{t-1}^\eta - \mathbf{u}_{t-1}\|^2 - \|\mathbf{x}_t^\eta - \mathbf{u}_t\|^2 + \|\mathbf{x}_{t-1}^\eta - \mathbf{u}_t\|^2 - \|\mathbf{x}_{t-1}^\eta - \mathbf{u}_{t-1}\|^2 - \|\mathbf{x}_t^\eta - \mathbf{x}_{t-1}^\eta\|^2 \right)$$
$$= \frac{1}{2\eta}\left( \|\mathbf{x}_{t-1}^\eta - \mathbf{u}_{t-1}\|^2 - \|\mathbf{x}_t^\eta - \mathbf{u}_t\|^2 + \langle \mathbf{x}_{t-1}^\eta - \mathbf{u}_t + \mathbf{x}_{t-1}^\eta - \mathbf{u}_{t-1}, \mathbf{u}_{t-1} - \mathbf{u}_t \rangle - \|\mathbf{x}_t^\eta - \mathbf{x}_{t-1}^\eta\|^2 \right)$$
$$\leq \frac{1}{2\eta}\left( \|\mathbf{x}_{t-1}^\eta - \mathbf{u}_{t-1}\|^2 - \|\mathbf{x}_t^\eta - \mathbf{u}_t\|^2 + \left( \|\mathbf{x}_{t-1}^\eta - \mathbf{u}_t\| + \|\mathbf{x}_{t-1}^\eta - \mathbf{u}_{t-1}\| \right)\|\mathbf{u}_t - \mathbf{u}_{t-1}\| \right)$$
$$\quad - \frac{1}{2\eta}\|\mathbf{x}_t^\eta - \mathbf{x}_{t-1}^\eta\|^2$$
$$\overset{(12)}{\leq} \frac{1}{2\eta}\left( \|\mathbf{x}_{t-1}^\eta - \mathbf{u}_{t-1}\|^2 - \|\mathbf{x}_t^\eta - \mathbf{u}_t\|^2 \right) + \frac{D}{\eta}\|\mathbf{u}_t - \mathbf{u}_{t-1}\| - \frac{1}{2\eta}\|\mathbf{x}_t^\eta - \mathbf{x}_{t-1}^\eta\|^2.$$

Summing the above inequality over all iterations, we have

$$\sum_{t=1}^T \left( f_t(\mathbf{x}_t^\eta) - f_t(\mathbf{u}_t) \right) \leq \frac{1}{2\eta}\|\mathbf{x}_0^\eta - \mathbf{u}_0\|_2^2 + \frac{D}{\eta}\sum_{t=1}^T \|\mathbf{u}_t - \mathbf{u}_{t-1}\| - \frac{1}{2\eta}\sum_{t=1}^T \|\mathbf{x}_t^\eta - \mathbf{x}_{t-1}^\eta\|^2$$

$$\overset{(12)}{\leq} \frac{1}{2\eta}D^2 + \frac{D}{\eta}\sum_{t=1}^T \|\mathbf{u}_t - \mathbf{u}_{t-1}\| - \frac{1}{2\eta}\sum_{t=1}^T \|\mathbf{x}_t^\eta - \mathbf{x}_{t-1}^\eta\|^2. \tag{68}$$

Then, the dynamic regret with switching cost can be upper bounded as follows

$$\sum_{t=1}^T \left( f_t(\mathbf{x}_t^\eta) + \|\mathbf{x}_t^\eta - \mathbf{x}_{t-1}^\eta\| - f_t(\mathbf{u}_t) \right)$$

$$\overset{(68)}{\leq} \frac{1}{2\eta}D^2 + \frac{D}{\eta}\sum_{t=1}^T \|\mathbf{u}_t - \mathbf{u}_{t-1}\| - \frac{1}{2\eta}\sum_{t=1}^T \|\mathbf{x}_t^\eta - \mathbf{x}_{t-1}^\eta\|^2 + \sum_{t=1}^T \|\mathbf{x}_t^\eta - \mathbf{x}_{t-1}^\eta\|$$

$$\leq \frac{1}{2\eta}D^2 + \frac{D}{\eta}\sum_{t=1}^T \|\mathbf{u}_t - \mathbf{u}_{t-1}\| - \frac{1}{2\eta}\sum_{t=1}^T \|\mathbf{x}_t^\eta - \mathbf{x}_{t-1}^\eta\|^2 + \sum_{t=1}^T \left( \frac{1}{2\eta}\|\mathbf{x}_t^\eta - \mathbf{x}_{t-1}^\eta\|^2 + \frac{\eta}{2} \right)$$

$$= \frac{1}{2\eta}D^2 + \frac{D}{\eta}\sum_{t=1}^T \|\mathbf{u}_t - \mathbf{u}_{t-1}\| + \frac{\eta T}{2}.$$