# OpenReview forum: "Revisiting Smoothed Online Learning"
_NeurIPS.cc/2021/Conference — NeurIPS 2021 Poster_

### Official Review · Reviewer_Njc7 · 2021-06-25

**Rating:** 6
**Confidence:** 3

**Summary:**

This paper studies smoothed online learning and considers two performance metrics: the competitive ratio and the dynamic regret. For the competitive ratio part, the authors study the greedy algorithm which minimizes the weighted sum of the hitting costs and the switching cost. They improve the existing competitive ratio results by some constant factors in three different settings. For the dynamic regret part, the authors show that a small modification of Ader, SAder, can attain the same dynamic regret bound as the case without switching costs. They also show that if the agent can observe next step's hitting cost before making decisions, the Lookahead SAder algorithm can achieve the optimal dynamic regret bound without the bounded gradient assumption.

Update: I partly agree with the points made in the response. So I adjust the score to 6.

**Ethical Concerns:**

I do not see any ethical issues with this paper.

**Limitations And Societal Impact:**

The two open problems at the end of the paper discuss the limitations of this work. I think they are appropriate.

Some constructive suggestions: I feel the authors can consider more about the applications of smoothed online learning, and use them to justify the assumptions they make on the hitting and switching costs. The switching costs may have more complicated forms than $\ell_2$ norm or $\ell_2$ norm squared, and the hitting costs may become much simpler (e.g. quadratic).

**Main Review:**

I have some concern with the originality of this work. Most algorithms considered in this paper have already been studied in similar settings, and the authors failed to convince me about the novelty of their proof techniques. I hope the authors can briefly clarify this in the rebuttal. Are these proofs trivial extensions of previous works? If not, what is the most significant challenge and how do you overcome it?

This paper is technically sound. All claims are supported by rigorous proofs. Although I did not check the details in the appendix, I think the results make sense.

The writing of this paper needs to be improved much. First, the overall organization is a bit messy. The authors present five different problem settings and analyze four different algorithms, but do not discuss much about their connections.  Second, the authors should focus more on the intuitions and novelties in their remarks, rather than only comparing the final bounds. I believe presentation issues can usually be fixed in the revision, but they may affect my evaluation of originality and significance.

My major concern is about the significance of this work. I hope the authors can help me understand the following issues better in the rebuttal:
1. I do not think the improvements in Thm 1 and Thm 2 are significant from a theoretical perspective. It is clear from the form of the competitive ratio that the problem becomes more challenging as $\alpha$ and $\lambda$ converges to zero. The new results in this paper are in the order of $O(1/\alpha)$ or $O(1/\lambda)$, which are the same as previous results. Since there are no lower bound results to rule out competitive ratios in the order of $O(1/\sqrt{\alpha})$, the presented results may still be far from optimal for small $\alpha$. Given this, the claim “It seems safe to ignore the switching cost” in line 211 looks weird to me.
2. In Section 3.3, the proposed algorithm is a special case of the ROBD algorithm in [Goel et al., 2019]. The problem setting becomes slightly more general in this paper, but the proof technique looks similar with the ROBD proof in [Goel et al., 2019]. I hope the authors can clarify why they think the new setting (convex + quadratic growth hitting costs) is important, and what is new in the proof.
3. In Section 4.1, the authors emphasize that “the only modification” they make to Ader is incorporating the switching costs. Does this lead to significant changes in the proof?

In summary, I suggest to reject this paper because I feel the contribution of this work is incremental, and the presentation is not good enough. I understand that the presentation issues may affect my evaluation of other aspects, so I am open to adjust the score if the authors can address some of my concerns.

**Time Spent Reviewing:**

6

---

> ### Author Response · Authors · 2021-08-07
> **Many thanks for the constructive reviews!**
>
> Many thanks for the constructive reviews! We will improve our paper according to the suggestions. We hope the reviewer could check our response, and reevaluate our paper. We are very happy to respond additional questions during the rolling discussion.
>
> ---
>
> Q1: I do not think the improvements in Thm 1 and Thm 2 are significant from a theoretical perspective.
>
> A1: We want to emphasize that when considering competitive ratio, the improvement on constant factors is *indeed* significant. Note that in the community of theoretical computer science, reducing the constant factor in competitive ratio is believed to be important. That is because competitive ratio implies *multiplicative* upper bounds, and thus the reduction in the constant factor leads to dramatic differences. For example, our Theorem 2 implies that the total loss is at most $(1+\frac{4}{\lambda}) \times Optimum$, and Lin et al. [2020] prove that their total loss is at most $\max(1 + \frac{6}{\lambda}, 4) \times Optimum$. Thus, our improvement when $\lambda < 2$ is
> $$
> \frac{(1+\frac{6}{\lambda})-(1+\frac{4}{\lambda})}{1 + \frac{6}{\lambda}} = \frac{2}{\lambda +6} \geq \frac{2}{2+6}= 25 \\%
> $$
> It is important to notice that the $25\\%$ improvement is *relative*, and thus significant. Furthermore, the improvement holds for all $\lambda < 2$.\
> For comparison, dynamic regret only implies *additive* upper bounds. In other words, the total loss is upper bounded by $Optimum + Dynamic \\ Regret $. Thus, constant factors in dynamic regret are not so important, and sometimes are ignored.
>
> ---
>
> Q2: Given this, the claim “It seems safe to ignore the switching cost” in line 211 looks weird to me.
>
> A2: The naïve approach that ignores the switching cost *indeed* achieves the smallest competitive ratio for polyhedral functions and quadratic growth functions. We also feel counterintuitive, and so we use the term “seems”. Furthermore, we have mentioned in Line 195 that “It is unclear whether this is an artifact of our analysis or an inherent property, and will be investigated in the future”.
>
> ---
>
> Q3: I hope the authors can clarify why they think the new setting (convex + quadratic growth hitting costs) is important.
>
> A3: First, convex and quadratic growth functions are more generally than strongly convex functions. Recently, quadratic growth functions have received significant attention from optimization and machine learning communities. Examples of quadratic growth functions that are non-strongly convex can be found in Section 4 of [Drusvyatskiy and Lewis, 2018] and Section 4 of Necoara et al. [2019]. \
> Second, Goel et al. [2019, Theorem 3] have also studied *quasiconvex and quadratic growth* functions, which are very close to *convex and quadratic growth* functions. By comparing the proof of our Theorem 3 with that of Goel et al. [2019, Theorem 3], we can see that our analysis is much more simple.
>
> ---
>
> Q4: The proof technique looks similar with the ROBD proof in [Goel et al., 2019]. What is new in the proof.
>
> A4: At the first glance, one may feel that our analysis of Theorem 3 looks similar with that of R-OBD [Goel et al., 2019, Theorem 4]. That is because both proofs use an amortized local competitiveness argument, and this is a standard technique in the analysis of competitive ratio [Bansal et al., 2015]. However, our proof is more challenging because the hitting cost could be non-strongly convex. The novelty of the proof lies in how to handle the two negative terms in Line 563
> $$
> -\frac{\gamma(\lambda + 4c) }{2\lambda} \|\mathbf{u}_t-\mathbf{x}_t\|^2 - c\|\mathbf{x} _{t-1}-\mathbf{u} _{t-1}\|^2
> $$
> The standard approach is to drop the first term, decompose the second term as
> $$
> \- c\|\mathbf{x} _{t-1}-\mathbf{u} _{t-1}\|^2 \leq - c \left(\frac{1}{1+\rho}\|\mathbf{x} _{t-1}-\mathbf{u} _t\|^2 -\frac{1}{\rho}\|\mathbf{u} _t - \mathbf{u} _{t-1}\|^2\right)
> $$
> and finally take summations over $t=1,\ldots, T$. However, it only yields an $O(1/\lambda)$ competitive ratio.
>
> To improve the competitive ratio, in Line 564, We first take summations over $t=1,\ldots, T$, and in this way, the two negative terms can be merged together and become
> $$
> -\left(\frac{\gamma(\lambda + 4c) }{2\lambda} +c \right)\sum_{t=1}^T \|\mathbf{x} _{t-1}-\mathbf{u} _{t-1}\|^2
> $$
> Then, we upper bound the above term by
> $$
> -\left(\frac{\gamma(\lambda + 4c) }{2\lambda} +c \right) \sum _{t=1}^T \left(\frac{1}{1+\rho}\|\mathbf{x} _{t-1}-\mathbf{u} _t\|^2 -\frac{1}{\rho}\|\mathbf{u} _t - \mathbf{u} _{t-1}\|^2\right).
> $$
> The rest of the proof is more or less standard. After careful analysis, we can prove a $1 + \frac{2}{\sqrt{\lambda}}$ ratio.
>
> ---
>
> Q5: In Section 4.1, the authors emphasize that “the only modification” they make to Ader is incorporating the switching costs. Does this lead to significant changes in the proof?
>
> A5: What we want to claim is that our algorithm is relatively simple. On the other hand, it is *nontrivial* to realize that one modification is sufficient. Specifically, we first analyze the challenge of bounding the dynamic regret with switching cost of the meta-algorithm, and from the theoretical result we find that it is necessary to incorporate the switching cost $\|\mathbf{x}_t^\eta - \mathbf{x} _{t-1}^\eta\|$ of expert $E^\eta$ to measure its performance. The analysis of the meta-regret of our SAder is significant different from that of Ader.\
> For details, we recommend the reviewer to check the proof of Lemma 1, which is the key technical contribution. We provide a novel decomposition of the meta-regret of SAder in (52), which motivates “the only modification”.
>
> ---
>
> Q6: Are these proofs trivial extensions of previous works? If not, what is the most significant challenge and how do you overcome it?
>
> A6: We would like to take this opportunity to emphasize our technical contributions.
>
> 1. In the proof of Theorem 1, our technical contribution is to make full use of the negative term to reduce the competitive ratio. Specifically, our treatment of $-\frac{2}{\alpha} f_t(\mathbf{x}_t) $ in (23) is different from Lin et al. [2020]. We agree that the proof is straightforward, but are surprised that such result did not appear in the literature. So, we feel it is worth to report this result as the baseline for future studies of polyhedral functions.
>
> 2. In the proof of Theorem 2, the challenge is how to determine the optimal value of $\rho$, and we propose a principled way to decide its value. We first characterize how the competitive ratio depends on $\rho$ in (27), and then identify the value of $\rho$ which minimizes the ratio.
> 3. The technical novelty of Theorem 3 is discussed in A4.
>
> 4. The technical novelty of Theorem 4 is discussed in A5.
>
> 5. The challenge of Theorem 5 is how to avoid the bounded gradient condition. The technical novelty is the analysis of the lookahead Hedge in Lemma 6 and the meta-regret in Lemma 3. The key is to keep the last negative term in (60), i.e.,
> $$
> \- \frac{1}{2\beta} \sum_{t=1}^T \|\mathbf{w}_t-\mathbf{w} _{t-1}\|_1^2,
> $$
> and use it to control the following term in the meta-regret
> $$
> D \sum _{t=2}^T  \|\mathbf{w}_t -\mathbf{w} _{t-1}\|_1,
> $$
> as shown in (61). In this way, the upper bound is independent from the norm of the gradient.
>
> 6. The lower bound in Theorem 6 is novel. In our opinion, it is very important because it shows that it is *impossible* to improve the $O(\sqrt{T(1+P_T)})$ upper bound even in the lookahead setting. The proof of Theorem 6 is based on a novel reduction of lower bound of dynamic regret to that of competitive ratio [Argue et al., 2020a].

---

> > ### Comment · Reviewer_Njc7 · 2021-08-16
> > **Thanks for the response!**
> >
> > I want to thank the authors for the detailed response. A2, A3, A5 addressed my concern. For A1, I agree that this result is worth reporting, but it is not significant enough. This is fine because the paper has many other results. For A4, I feel the technique the authors mentioned is very good, but it is hard to be summarized as a novel proof idea. However, the whole proof is a good contribution because it is simpler than the proof of Thm 3 in [Goel et al. 2019] and the constants are exactly specified. I still feel the authors need to make more concrete connections among all the algorithms (with different parameters) and problem settings. The current presentation looks more like a summary of irrelevant results rather than a complete story. It is difficult for me to provide any specific suggestion about this issue.
> >
> > I will adjust the score to 6. I encourage the authors to add some of their response into the main body in revision.

---

> > > ### Author Response · Authors · 2021-08-17
> > > **Many thanks! We will revise our paper accordingly.**
> > >
> > > Dear Reviewer Njc7,
> > >
> > > Thank you very much for your kind reply! We will revise our paper according to the suggestions.
> > >
> > > Best\
> > > Authors

---

### Official Review · Reviewer_C2AR · 2021-07-13

**Rating:** 6
**Confidence:** 4

**Summary:**

The authors study online convex optimization with switching costs, a well-known problem which has attracted much recent attention in the past few years, and which is closely related to convex body chasing. The authors make the following two contributions: first, they improve on some of the competitive ratio bounds presented in Goel et al and Lin et al, and second, they present an algorithm with optimal dynamic regret, along with a matching lower bound.

**Ethical Concerns:**

No problems here.

**Limitations And Societal Impact:**

No problems here.

**Main Review:**

Theorem 1 claims to improve on the competitive ratio presented in Lin et al, but the improvement is minor and the proof is trivial. I do not believe this theorem will be of interest to the broader ML community. Theorems 2 and 3 are more interesting, and use a careful analysis to simplify and improve results from Goel et al. The proofs are simple and clean, and they are also able to match a lower bound proved in Goel et al. It's nice to see the exact constants nailed down. Theorems 4 and 5 prove dynamic regret bounds for a modified version of the Ader algorithm introduced in Zhang et al; Theorem 6 proves a matching lower bound. The algorithm they propose is only a slight modification of vanilla Ader, so I think the main contribution here is the analysis and the new lower bound.

I do have a suggestion for the authors - they write on line 141 that the squared L2 distance considered in the dynamic regret bounds of Goel et al is "not suitable for general convex functions." It seems to me that the point the authors are making is that the dynamic regret bounds proved in Goel et al rely on strong convexity, whereas their regret bounds do not. While this is true, it does not imply that L2 switching cost is always a superior metric to squared L2 switching cost - different metrics are more relevant for different problems. One advantage of the squared L2 distance is that it is related to LQR control problems, as shown by Goel and Wierman. I think it would be helpful to add a paragrpah explaining the different roles that different choices of switching cost play, instead of writing as if one choice is strictly superior to other choices.

Overall opinion: while the paper does not provide any new algorithmic insights per se, it does present small improvements on existing results on a problem that has attracted much recent attention in the ML community. The paper is well-written and acknowledges previous work properly.

**Time Spent Reviewing:**

2

---

> ### Author Response · Authors · 2021-08-07
> **Many thanks for the constructive reviews!**
>
> Many thanks for the constructive reviews!
>
> ---
>
> Q1: Theorem 1 claims to improve on the competitive ratio presented in Lin et al, but the improvement is minor and the proof is trivial.
>
> A1: We agree with the reviewer that the analysis of Theorem 1 is straightforward. We are very surprised that such result did not appear in the literature, so we feel it is worth to report this result as the baseline for future studies of polyhedral functions.
>
> ---
>
> Q2: I do have a suggestion for the authors - they write on line 141 that the squared L2 distance considered in the dynamic regret bounds of Goel et al is "not suitable for general convex functions." … I think it would be helpful to add a paragraph explaining the different roles that different choices of switching cost play, instead of writing as if one choice is strictly superior to other choices.
>
> A2: Thanks for the suggestion. We totally agree with the reviewer that different metrics are more relevant for different problems. What we want to claim is that when considering general convex functions, it is better to use the $\ell_2$-norm as the switching cost, and when facing strongly convex functions (or quadratic functions), it is better to use the squared $\ell_2$-norm as the switching cost. In this way, we can guarantee that the hitting cost and the switching cost are roughly on the same order. We will revise our paper to make it more clear.
>
> ---
>
> Q3: It seems to me that the point the authors are making is that the dynamic regret bounds proved in Goel et al rely on strong convexity, whereas their regret bounds do not.
>
> A3: If we check their analysis, we can see that the dynamic regret proved by Goel et al. [2019, Theorem 6] *didnot* rely on strong convexity. In fact, the upper bound in (47b) of their proof is minimal when the strong convexity parameter $m=0$. \
> To be precise, when the hitting cost is general convex, Goel et al. [2019] prove an upper bound for dynamic regret with squared $\ell_2$-norm switching cost. Of course, they can use the same bound when the hitting cost is strongly convex.

---

> > ### Comment · Reviewer_C2AR · 2021-08-29
> > **Thanks for the response!**
> >
> > After reading the other review's and the author's responses, I feel comfortable with my review and am leaving my score unchanged.

---

### Official Review · Reviewer_MZJx · 2021-07-17

**Rating:** 6
**Confidence:** 4

**Summary:**

This paper studies smoothed online learning by targeting two performance metrics: competitive ratio and dynamic regret with switching cost. The paper first improves the competitive ratio of the naive approach and the greedy algorithm. Then the paper proposes a novel algorithm called SAder which achieves optimal dynamic regret with switching cost under the lookahead setting.


**Limitations And Societal Impact:**

See main review

**Main Review:**

Strengths:

The contribution of this paper is solid. It contains two parts. The first part (section 3) provides better competitive ratios for the naive approach and the greedy algorithm. The second part (section 4) includes a novel algorithm with optimal dynamic regret under the lookahead setting.

Weakness:

1. These two parts of the contribution are independent of each other to some extent. Is it possible to study one algorithm which can simultaneously improve both competitive ratio and the dynamic regret? Notice that the OBD and ROBD algorithms have theoretical guarantees on both competitive ratio and the dynamic regret, though their results are not optimal.

2. The paper does not contain the lower bound of polyhedral functions and quadratic growth functions.

**Time Spent Reviewing:**

6

---

> ### Author Response · Authors · 2021-08-07
> **Many thanks for the constructive reviews!**
>
> Many thanks for the constructive reviews!
>
> ---
>
> Q: Is it possible to study one algorithm which can simultaneously improve both competitive ratio and the dynamic regret?
>
> A: There exists a general strategy to improve both the competitive ratio and the dynamic regret. We first run two different algorithms to minimize the competitive ratio and the dynamic regret *separately*, and then use the meta-algorithm of Daniely and Mansour [2019] to aggregate the two algorithms. In this way, we can achieve the best of both worlds, i.e., improving both competitive ratio and dynamic regret.

---

> > ### Comment · Reviewer_MZJx · 2021-08-31
> > **Reply**
> >
> > After reading authors' response and other reviews, I will keep my score. I suggest incorporating the connection between two parts of contribution to the revision.

---

> > > ### Author Response · Authors · 2021-08-31
> > > **Thanks for the suggestion!**
> > >
> > > Dear Reviewer MZJx,
> > >
> > > Thank you very much for your kind reply! We will revise our paper according to the suggestion.
> > >
> > > Best\
> > > Authors

---

### Official Review · Reviewer_A46c · 2021-08-01

**Rating:** 6
**Confidence:** 3

**Summary:**

This work considers the problem of online learning with movement cost, and studies the metrics of competitive ratio and dynamic regret for polyhedral and quadratically-growing functions. For competitive ratio, while studying simple algorithms (a positive) that ignore switching costs, the upper bounds offered here improve over the best known in both settings (for distinct switching cost in each case). For dynamic regret, the paper suggests a modification of a dynamic-regret-guaranteeing algorithm to the switching cost setting with optimal dynamic regret.

**Limitations And Societal Impact:**

Please see above.

**Main Review:**

On the competitive ratio front, the paper presents concrete improvements for special cases of polyhedral and quadratically growing functions (~strong convexity around minimum).

As for dynamic regret: It is well-known that OGD gets the correct dynamic regret as long as the step size choice is allowed to depend on the path length of the comparator (an oracle quantity). Therefore, existing algorithms essentially run Mult Weights over guesses of choices of exponentially spaced learning rates. The modification here for accomodating movement cost is to penalize the experts (corresponding to learning rate guesses) for not just their decision, but also for their movement. This is natural choice.

Questions:
1. Previous work in this setting could operate on locally polyhedral functions. Can such functions be accomodated here? If not, what is the precise obstruction to this?
2. What is the claim of novelty in section 3.3? In the reviewer's reading, the upper bound in Goel et al (Beyond OBD) matches the lower bound exactly for squared l2 movement (see discussion below Theorem 4 there). Theorem 3 here seems to offer strictly worse bounds, contradicting "the constants in our bound are much smaller."

**Time Spent Reviewing:**

4

---

> ### Author Response · Authors · 2021-08-07
> **Many thanks for the constructive reviews!**
>
> Many thanks for the constructive reviews! There are some misunderstandings about previous studies, which are clarified below. We hope the reviewer could check our response, and reevaluate our paper. If necessary, we are very happy to answer more questions during the rolling discussion.
>
> ---
> Q1: Previous work in this setting could operate on locally polyhedral functions. Can such functions be accommodated here? If not, what is the precise obstruction to this?
>
> A1: In fact, previous work *cannot* operate on locally polyhedral functions. Although Chen et al. [2018, Definition 6] define locally $\alpha$-polyhedral functions, they never use that definition. Throughout their paper, Chen et al. [2018] actually use the global polyhedral functions as our paper. To the best of our knowledge, no previous studies really consider locally polyhedral functions.\
> It is difficult to use the concept of “locally polyhedral functions”, because we do not have any control over the distance among the minimizers of different hitting costs. If we assume the minimizers of successive hitting costs are close, then it may be possible to operate on locally polyhedral functions.
>
> ---
> Q2: What is the claim of novelty in section 3.3? In the reviewer's reading, the upper bound in Goel et al (Beyond OBD) matches the lower bound exactly for squared l2 movement (see discussion below Theorem 4 there). Theorem 3 here seems to offer strictly worse bounds, contradicting "the constants in our bound are much smaller.
>
> A2: Our Theorem 3 is designed for *convex and quadratic growth* functions, and Theorem 4 of Goel et al. [2019] is designed for *strongly convex* functions. So, they are not directly comparable, because convex and quadratic growth functions could be non-strongly convex. \
> On the other hand, since convex and quadratic growth functions are more general than strongly convex functions, the lower bound of the former one is larger than the lower bound of the latter one. So, we can still use the lower bound of strongly convex functions in Theorem 1 of Goel et al. [2019] to show that our Theorem 3 is optimal up to constant factors.\
> Theorem 3 of Goel et al. [2019] is designed for *quasiconvex and quadratic growth* functions, which are very close to *convex and quadratic growth* functions. So, we should compare our Theorem 3 with Theorem 3 of Goel et al. [2019]. In this case, our constant is much smaller and our proof is much simpler. Note that Goel et al. [2019] did not provide the exact constant in their Theorem 3, but from their analysis it is easy to see that the constant is very large.

---

> > ### Author Response · Authors · 2021-08-25
> > **Could you please check our response?**
> >
> > Dear Reviewer A46c,
> >
> > Could you please check our response to see if we address your concerns? In our opinion, the two questions are caused by the unfamiliarity with previous results, and clarified in the response.
> >
> > Many thanks for your time!
> >
> > Best\
> > Authors

---

> > > ### Comment · Reviewer_A46c · 2021-08-30
> > > **Revised score**
> > >
> > > Thanks. The clarification was satisfactory on both points raised.
> > >
> > > I have accordingly revised my score.

---

> > > > ### Author Response · Authors · 2021-08-31
> > > > **Many thanks! We will improve our paper accordingly.**
> > > >
> > > > Dear Reviewer A46c,
> > > >
> > > > Thank you very much for your kind reply! We will improve our paper according to your suggestions.
> > > >
> > > > Best\
> > > > Authors

---

### Decision · Program_Chairs · 2021-09-27

**Decision:**

Accept (Poster)

**Comment:**

The paper considers the analysis of competitive ratio/dynamic regret for problems with hitting cost and switching cost. They provide concrete improvements in terms of nailing down the constants for the ratios in the case of polyhedral function and quadratic growth functions achieving this with a relatively simple algorithm and a clean analysis. The reviewers have appreciated the clarity and simplicity of the paper and the approach, the concreteness of the improvements achieved as well as the right attribution and discussion of previous work. Nevertheless, the paper does not provide any novel algorithmic insights and analyses a well-known algorithm in various settings with improvements in guarantees. Overall while the paper is close to the borderline post the discussion, the reviewers unanimously agreed that the paper is slightly above the borderline due to the concrete improvements for well-known and well-studied problems and there by I am proposing an accept.